# An explainable deep learning model based on hydrological principles for flood simulation and forecasting

Xin Xiang[1], Shenglian Guo[1,*], Chenglong Li[1], Yun Wang[2]

[1]State Key Laboratory of Water Resources Engineering and Management, Wuhan University, Wuhan, 430072, China

[2]Water Resources Technical College, Wuhan, 430072, China

*Correspondence to: Shenglian Guo (slguo@whu.edu.cn)*

**Abstract:** Deep learning (DL) models always perform well in hydrological simulation but lack physical-based principles. To address this gap, we integrate the relatively complex runoff generation and flow routing principals of Xinanjiang (XAJ) model into the architecture of recurrent neural network (RNN) units and establish a physical-based XAJRNN layer. Subsequently, this layer is fused with LSTM layers to construct an explainable deep learning (EDL) model, which underwent testing at the Lushui River and Qingjiang River basins in China. Compared to benchmark models, the proposed EDL model performs very well, the average Nash-Sutcliffe efficiency (*NSE*)values for these two basins are 0.98 and 0.94, respectively. The flood peak relative errors (*PRE*) and peak timing difference ($\Delta T$) are close to zero, which demonstrate that the EDL model can accurately simulate flood events. Notably, the EDL model incorporated physical principles not only can improve flow simulation accuracy, but also enhance interpretability, which offer fresh insights for the fusion of DL and hydrological models for flood simulation and forecasting.

## 1 Introduction

In the modern era, flood disasters present substantial threats to both human societies and the natural environment (Guido et al., 2023). With the intensification of global climate change and rapid urbanization, the accuracy and timeliness of flood forecasting have become increasingly important. Flood forecasting typically relies on hydrological models (Thaisiam et al., 2024). By analyzing the rainfall-runoff relationships from historical periods, hydrological models simulate hydrological processes within

a basin. Combining these with forecasted rainfall, such models can forecast flow discharges, water levels,
and probabilities of flood occurrence. In recent years, advancements in computational power and
artificial intelligence (AI) technologies have significantly improved the accuracy and real-time
responsiveness of hydrological models (Hirabayashi et al., 2013), offering more scientific and efficient
support for disaster prevention and mitigation efforts.
Traditional hydrological models rely on statistical methods and empirical formulas but struggle to
accurately simulate complex nonlinear hydrological processes (Roy et al., 2023). Their predefined
equations are unable to adapt the climate and environmental changes such as land use and human
activities. Additionally, these models often simplify rainfall spatiotemporal distribution and the land
surface heterogeneity. In recent years, deep learning (DL) technologies have made significant
advancements in various fields, particularly in time series prediction, where they display strong potential.
DL, a domain dedicated to uncovering patterns and extracting knowledge from large datasets, enables
computers to autonomously learn algorithms, analyze extensive sample data, and identify patterns,
facilitating predictions on unfamiliar data. This process closely aligns with hydrological modeling, which
discerns patterns by analyzing historical hydrometeorological data, generalizing, and simulating
hydrological processes. Consequently, DL has attracted widespread attention in hydrology (Nearing et
al., 2021).
DL commonly refers to deep neural networks, a form of representational learning technique that
links simple nonlinear computational units through multi-layer network architectures to understand
intricate relationships. It falls within the realm of machine learning (ML) (Yann et al., 2015). The term
"deep" in DL signifies network structures with multiple layers and neurons, although there is no precise
definition of "deep". Generally, it denotes models that necessitate substantial data and encompass
numerous layers and neurons. These layers convert their inputs into higher-level features, magnifying
crucial factors for output variability while reducing irrelevant variations. This facilitates automatic
feature extraction, contrasting with "shallow" networks or conventional ML algorithms that rely on
expert knowledge and engineering skills for designing feature extractors. This is a key rationale behind
the increasing application of DL models over shallow networks in recent years (Frank et al., 2020).
Structurally, the standard recurrent neural network (RNN), exemplified by long short-term memory

(LSTM), remains the foundational model architecture for DL-driven hydrological forecasting. As a subset of RNN in DL, LSTM has gained prominence for its efficacy in managing sequential data and capturing long-term dependencies. LSTM tackles the challenges of vanishing and exploding gradients in traditional RNNs when handling lengthy sequences through gated mechanisms, resulting in superior performance in time-dependent prediction tasks (Hochreiter and Schmidhuber, 1997).

DL has been extensively utilized in various fields. In hydrology, where processes are not yet fully understood, it exhibits promise in identifying physical processes through a data-mining lens. However, achieving accurate forecasting is not the sole aim. Hydrologists are interested in whether models are in line with fundamental physical principles, are interpretable, and contribute to scientific knowledge advancement. Traditional physics-based hydrological models generally provide better interpretability and physical consistency, relying less on data and complementing DL models. As a result, the fusion of physics-based mechanisms and data-driven models has garnered significant attention in recent years, showcasing potential in advancing scientific inquiry (Nearing et al., 2021; Shen, 2018). Currently, the coupling of DL and physics-based models focuses on four main aspects.

(1) Introducing physical mechanisms into DL models' loss functions

Worland et al. (2019) developed a multi-output multilayer perceptron (MLP) model to forecast flow duration curve (FDC) quantiles, incorporating FDC monotonicity constraints into the loss function. This approach resulted in forecasts that adhered to monotonicity and closely matched the FDC derived from observations. Wang et al. (2020) not only incorporated physical laws into the loss function but also integrated expert knowledge in the form of inequalities, constructing theory-guided neural networks (TGNN). TGNN demonstrated superior predictive performance compared to standard DL models. Xie et al. (2021) encoded three physical conditions in rainfall-runoff forecasting into the loss function. Experiments across 531 Catchment Attributes and Meteorology for Large-sample Studies (CAMELS) basins showed improvements in the average Nash-Sutcliffe efficiency ($NSE$) from 0.52 to 0.61, enhanced peak flow simulating, and reduced unreasonable negative values. Pokharel et al. (2023) tested the effects of incorporating mass balance, energy balance, and storage-discharge relationships into the loss function of DL models across 34 basins, finding performance improvements in some basins, particularly with mass and energy balance constraints, which were effective in 38% and 32% of basins, respectively. Frame

et al. (2023) concluded that strictly adhering to the principle of water balance may reduce forecasting
accuracy due to data errors. However, DL models do not require the enforcement of the water balance
principle and can adapt to data biases, and they perform better than the traditional hydrological models
in flow forecasting.
(2) Using DL models as post-processors
Correcting errors in forecasting from physics-based models can significantly improve forecasting
accuracy. Cho and Kim (2022) employed LSTM to learn correlations between meteorological data and
WRF-Hydro forecast errors, applying this approach to calibrate WRF-Hydro forecasting. Experiments
in South Korean basins showed *NSE* values reaching 0.95, compared to 0.72 before calibration. Similarly,
Han and Morrison (2022) and Frame et al. (2021) applied LSTM to post-process multi-period forecasting
from the National Water Model (NWM) in the United States. Boucher et al. (2020) utilized the simulated
runoff of the GR4J hydrological model and observed water temperature as inputs to construct an MLP
model, demonstrating notable improvements compared to models without assimilation. Cui et al. (2021)
proposed a novel hybrid model combining the Xinanjiang (XAJ) hydrological model with LSTM for
multi-step flood forecasting. This model used XAJ outputs as inputs to the LSTM, enhancing the physical
mechanism of DL models. Yang et al. (2020) proposed a hybrid modeling framework that integrates a
physically distributed hydrological model (GBHM), artificial neural networks (ANN), a categorization
approach (CA), and computer vision (CV) to enhance hydrological simulations in data-scarce watersheds.
They show that these models can significantly improve the ability to capture spatial variability and
simulate extreme flow events. Li et al. (2014) proposed a black-box model which combines the back-
propagation neural network (BPNN) with the K-nearest neighbor (KNN) algorithm, and developed two
hybrid models (XBK and XSBK) by coupling the black-box routing module with the runoff generation
and separation modules of the XAJ model. Applications in multiple watersheds demonstrate that these
hybrid models outperform both the traditional BPNN and the XAJ model.
(3) Using DL models to calibrate parameters in traditional hydrological models
Tsai et al. (2021) proposed a parameter learning method to calibrate HBV model parameters. The
DL model generated parameters instead of directly outputting runoff, which were then combined with
inputs to produce runoff through the physical model. Applying this method across 1,802 basins showed
median Kling-Gupta Efficiency (*KGE*) values improving from 0.48 to 0.59 compared to calibration via
evolutionary algorithms followed by parameter regionalization. Similarly, Feng et al. (2023, 2022), Shen
et al. (2023) and Song et al. (2024) used DL models to calibrate HBV model parameters based on
meteorological data and basin attributes, driving hydrological models to simulate runoff. In addition to
the above calibration of lumped model parameters, a similar method is also used to calibrate the routing
model parameters. Zhong et al. (2024a, 2024b) used DL model to calibrate parameters in the Muskingum-
Cunge method and construct a distributed physics-driven DL hydrological model. Bindas et al. (2024)
introduced a novel differentiable routing method (δMC-JuniatahydroDL2) combining the Muskingum-
Cunge routing model with neural networks to infer Manning's roughness and channel geometry
parameters. The method provided more accurate long-term routing forecasts, especially in untrained sub-
basins.

(4) Designing DL models based on physical mechanisms

Encoding rules directly into neural networks represents a direct fusion of physics-based and data-

driven models. Hoedt et al. (2021) modified LSTM structures to enforce water balance over specific
periods. Experiments across 531 CAMELS basins showed improved peak flow performance despite no
overall improvement in *NSE*. Wang and Gupta (2024) explored the use of mass-conserving perceptron
(MCP)-based directed graph architectures to develop minimal and interpretable hydrological model
structures within a single catchment. They found that this framework significantly enhances flow
simulation performance, particularly when augmented with input bypass and bi-directional groundwater
exchange mechanisms. Jiang et al. (2020) modified RNN structures to incorporate state variables (e.g.,
soil moisture) from EXP-HYDRO model as recurrent unit states, combining these with other neural
network layers to construct a physics-guided RNN. Experiments in 671 CAMELS basins demonstrated
median NSE improvements from 0.60 to 0.71, with reductions in peak flow bias and improved baseflow
simulations. De la Fuente et al. (2024) proposed HydroLSTM, which models hydrological principles to
enhance interpretability, achieving comparable performance to LSTM models while requiring fewer unit
states. Similarly, Li et al. (2024) embedded EXP-HYDRO processes into RNN units, developing a
process-driven DL model that enhanced process understanding of rainfall-runoff relationships.
Experiments across 531 CAMELS basins demonstrated improvements over LSTM model. Similarly, He

et al. (2024) proposed a deep process learning (DPL) approach, which allows neural networks to infer underlying process mechanisms from observational data by embedding intuitive physical laws of geosystems directly into the DL architecture as structural priors. Wang et al. (2024) introduced a novel distributed hydrological modeling framework combining HydroPy model principles encoded into RNN units and DL models to calibrate physical parameters, improving runoff and water volume simulation performance

Currently, the current integration of DL models with physical mechanisms mainly involves loosely coupled approaches, such as modifying loss functions or calibrating parameters. Even the more advanced methods which embed physical mechanisms directly into neural network layers, they are relied on relatively simple or empirical physical models. To achieve real breakthroughs in hydrological forecasting, it is still necessary to systematically integrate more complex hydrological physical processes into neural network architectures, thereby ensuring both rigorous physical interpretability and superior forecasting performance under future scenarios.

The XAJ model proposed by Zhao (1992, 1993) has been widely used for hydrological simulation and flood forecasting in China. The flow routing of the XAJ model includes hillslope routing and channel network routing (Yao et al., 2009, 2014), which are represented by linear reservoirs and Nash unit hydrographs (Singh, 1977), respectively. Compared with other lumped hydrological models, the XAJ model performs very well in the humid and semi-humid regions, which helps to better highlight the distinctive aspects of this study. The core innovation of this study is the development of a novel XAJRNN layer that converts the XAJ model's sophisticated runoff generation and flow routing mechanisms into differential equation form and embeds them within a conventional RNN unit framework by explicitly defining its state variables and fluxes. An explainable deep learning (EDL) model combining the XAJRNN layer and LSTM layer is constructed and tested in the Lushui River and Qingjiang River basins to demonstrate the advantages of the EDL model in flood simulation. The findings may offer a promising new avenue for tightly integrating complex hydrological processes into DL models to improve flood forecasting accuracy.

The rest of this paper is organized as follows. The case study and materials are introduced in Section 2. Section 3 presents the methodologies. Section 4 evaluates and analyses the simulated results. Section

5 discusses the strengths as well as the weaknesses of the proposed model. Conclusions and outlook are

given in Section 6.

**2 Study area and data**

**2.1 Study basin**

**(1) Lushui River basin**

The Lushui River is a primary tributary of the middle Yangtze River, with a basin area of approximately 3,950 km². The basin's terrain slopes from high elevations in the southeast to lower areas in the northwest. The basin is located in a subtropical monsoon climate zone, characterized by warm and humid conditions, with an average annual temperature of approximately 15.5°C and an average annual rainfall of 1,550 mm. The Lushui River's annual runoff volume reaches 3.03 billion m³, with rainfall concentrated from May to September, accounting for 70% of the annual total. At the river valley's outlet, the Lushui Reservoir has a total storage capacity of approximately 408 million m³, with only 163 million m³ allocated for flood control. In early July 1995, the reservoir experienced its largest flood event, with an inflow peak of 4,500 m³/s and a three-day runoff of 500 million m³. In 2017, six flood events occurred, with peak inflows exceeding 1,000 m³/s, reaching a maximum inflow of 4,400 m³/s (Cui et al., 2021; Xiang et al., 2024). The geographical location of Lushui Reservoir is shown in Figure 1.

**(2) Qingjiang River basin**

The Qingjiang River, a primary tributary of the Yangtze River in the middle reaches, has a basin area of approximately 17,000 km². The region receives an average annual rainfall of 1,460 mm, with the majority falling between April and September, representing 75%–78% of the annual total. Situated in the heavy rainfall region of western Hubei Province, the basin's terrain facilitates the uplift of warm, moist air and is frequently affected by southwest cyclones. With a natural elevation drop of 1,430 m, the area features steep terrain and a high river gradient, leading to swift water flow convergence and significant fluctuations in flood levels. Consequently, the area is susceptible to severe rainfall and flood disasters. Along the main stream of the Qingjiang River, three sizable reservoirs exist, with the Shuibuya Reservoir serving as the central hub for the basin's cascade development, overseeing an area of roughly 10,860 km².

Located in Badong County, Hubei Province, the Shuibuya Reservoir plays a crucial role in the
hydropower development of the Qingjiang River. The Qingjiang River basin experienced major floods
in 2016 and 2017, with peak inflow discharge of the Shuibuya Reservoir reached 13,100 m³/s and 6,710
m³/s, respectively. It greatly forms an integral component of the flood control system in the middle and
lower reaches of the Yangtze River (Zhou et al., 2014). This study specifically focuses on the basin
controlled by the Shuibuya Reservoir, as depicted in Figure 1.

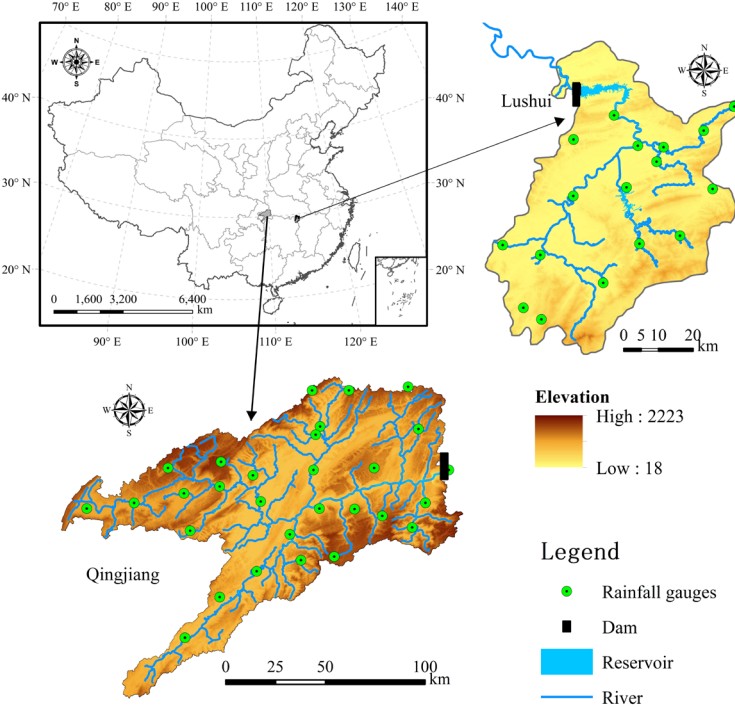

**Figure 1: Sketch map of river networks and rainfall gauges in the Lushui River and Qingjiang River basins.**
**2.2 Data**
The study collected flood season data (Lushui River basin: May 1 to October 31, 2012–2019;
Qingjiang River basin: April 1 to October 31, 2012–2020) that includes rainfall, pan evaporation, and
inflow datasets. It should be noted that the time step of these data series is 3 h in the Lushui River basin,
whereas it is 6 h in the Qingjiang River basin. For Lushui River basin, 3 h rainfall data from 17 gauges,
3 h pan evaporation, and 3 h inflow discharge was collected. The data from 2012 to 2016 were used for
training, and the data from 2017 to 2019 for testing. For Qingjiang River basin, 6 h rainfall from 28
gauges, 6 h pan evaporation, and 6 h inflow discharge was gathered. The data from 2012 to 2016 were
used for training, and the data from 2017 to 2020 for testing. The Thiessen polygon method was used to
calculate the areal mean rainfall and pan evaporation for both basins.

**3 Methodologies**

**3.1 XAJRNN neural network layer**

**3.1.1 XAJ model overview**

Zhao (1992, 1993) firstly proposed the XAJ model for rainfall-runoff simulation and flood
forecasting in the 1970s, and it is a classic conceptual hydrological model and has been widely used in
China. The core concept of the XAJ model is the runoff formation on repletion of storage: runoff is not
produced until the soil moisture content of the aeration zone reaches the field capacity. Once this
threshold is exceeded, all additional rainfall is converted directly into runoff without further loss. Runoff
production at a specific point occurs only after the tension water storage at that point is fully saturated.
To represent the spatial heterogeneity of tension water capacity within a basin, the model introduces a
tension water capacity curve. In terms of runoff separation, the model, based on empirical observations
and theoretical studies, adds an additional component: interflow, on top of the original division into
surface runoff and groundwater runoff (Yao et al., 2009, 2014). In this study, the evapotranspiration of
the XAJ model uses a three-layer soil moisture model. The runoff generation uses a tension water
capacity curve (Zhao, 1992, 1993). In the runoff separation, the runoff is divided into three types using
the free water capacity curve: surface runoff, interflow runoff, and groundwater runoff. The flow routing
includes both hillslope routing and channel network routing submodules, which use linear reservoirs and
Nash unit hydrographs (Singh, 1977), respectively. The logical structure of the XAJ model is shown in
Figure 2.
The model consists of input variables, state variables, fluxes, output variables, and parameters, along
with their corresponding mathematical equations. Input variables include areal mean rainfall and
measured pan evaporation, while the output variable is the simulated runoff. State variables represent
physical quantities that characterize the basin's state, and their dimensions are independent of time (La
Follette et al., 2021). The state variables of the XAJ model are shown in Table A1. Fluxes describe the
exchange of water within the basin or between the basin and external environments. These can be

expressed as functions of state variables or other fluxes, and their dimensions are time-dependent (La Follette et al., 2021). The fluxes of the XAJ model are shown in Table A2. The mathematical equations can be divided into state variable control equations and constitutive equations for fluxes. The control equations describe how state variables evolve over time, while the constitutive equations establish the relationships between unknown fluxes and state variables or known fluxes. Detailed information on the XAJ model is provided in Text A1.

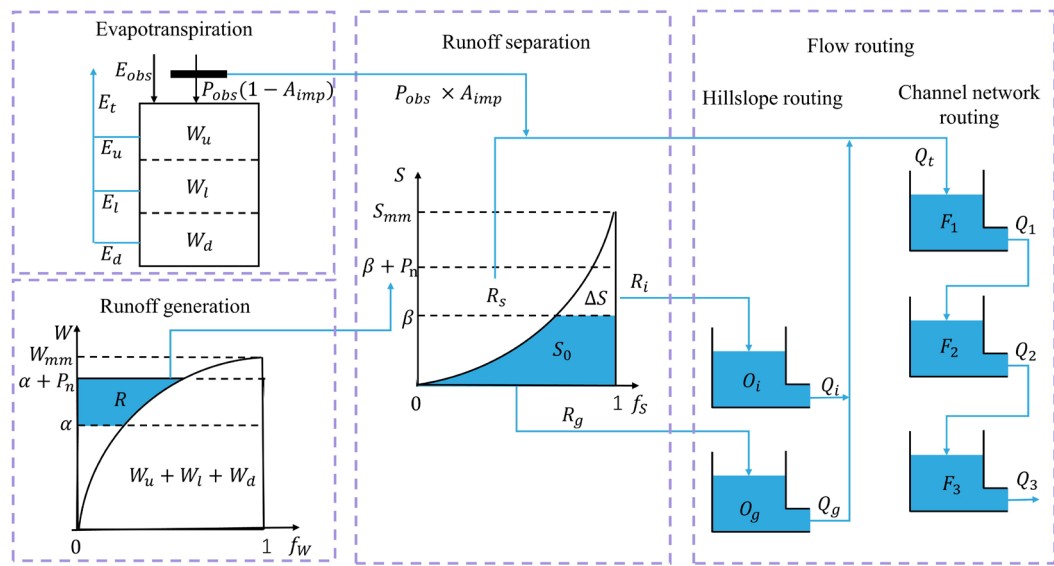

**Figure 2: Structure diagram of the XAJ model.**

**3.1.2 Derivation of XAJRNN**

Establishing the XAJ model in the watershed can be considered a complete system that represents changes in state variables within the watershed, such as the variation in the average tension water storage of the upper soil layer. At the same time, the XAJ model also describes how the watershed system responds to specific input conditions. These responses can be expressed through a combination of ordinary differential equations (ODE) and output equations:

$$\begin{cases} \frac{d}{dt}h(t) = F(h(t), x(t); \varphi_f) \\ y(t) = G(h(t), x(t); \varphi_g) \end{cases} \quad (1)$$

where: $h(t)$ represents the state variables of the XAJ model (as shown in Table 1). $x(t)$ represents the input variables of the XAJ model ($P_{obs}$ and $E_{obs}$). $y(t)$ represents the output flow of the XAJ model. $\varphi_f$ and $\varphi_g$ are parameters in the XAJ model (as shown in Table A3). $F(\cdot)$ and $G(\cdot)$ represent the

mathematical equations and functions in the model. The above equations form explicit continuous
equations, but in practice, implicit discrete equations are generally used to obtain numerical solutions:
$$\begin{cases} h(t) = f(h(t-1), x(t); \varphi_f) \\ \quad y(t) = g(h(t), x(t); \varphi_g) \end{cases} \quad (2)$$

Ordinary RNN is neural network structures specifically designed for handling sequential data, as

shown in Figure 3(a). RNN utilizes time-dependent relationships in sequences by storing previous state
information to assist in current computations (Rumelhart et al., 1986). At $t$th time step, the calculation
in an RNN unit can be divided into two steps: the first step is updating the hidden state ($h_t$), and the
second step is calculating the output ($y_t$). The calculation formulas are as follows:
$$\begin{cases} h_t = \sigma(W_{xh} \cdot x_t + W_{hh} \cdot h_{t-1} + b_h) \\ \quad y_t = \sigma(W_{hy} \cdot h_t + b_y) \end{cases} \quad (3)$$

where, $h_t$, $x_t$ and $y_t$ are the state, input, and output at $t$th time, respectively. $W_{xh}$, $W_{hh}$ and $W_{hy}$
are the weight parameters. $b_h$ and $b_y$ are bias parameters. $\sigma$ is the nonlinear activation function.

It can be observed that Eq. (2) and (3) have a similar structure. Both equations consist of two parts:

an ordinary differential equation and an output equation, and they share a highly similar structure.
Specifically, in the ordinary differential equation part, both equations include the state variable from
the previous time step ($h$(t-1)), the state variable at the current time step ($h$(t)), the input ($x$), and the
parameters (($\varphi$, $W$, $b$)). In the output equation part, both equations rely on the current state variable
($h$(t)), the output ($y$), and the same set of parameters (($\varphi$, $W$, $b$)). Therefore, in this study, we modify
the ordinary RNN unit structure by replacing the original equations and parameters with those derived
from the XAJ model, resulting in the XAJRNN. Similar to the ordinary RNN structure (Rumelhart et al.,
1986), the backbone of the XAJRNN layer consists of recurrent units that provide memory of past
sequences. In the XAJRNN layer structure, the connections between the recurrent units are represented
by implicit discrete equations (Eq. 3), and the parameters (i.e., weight parameters and bias parameters)
in the ordinary RNN unit are replaced by the physically meaningful parameters (as depict in Table A3)
from the XAJ model. Niu et al. (2019) demonstrated the connection between RNN network architecture
and numerical methods for ODE, theoretically supporting the use of XAJRNN for solving the dynamics
of the XAJ model system. Table A4 summarizes the pseudocode for implementing the XAJRNN unit.

The internal computation process of the XAJRNN unit is explained below using pseudocode and

equations provided in the appendix. For each XAJRNN unit, it is first necessary to initialize the
watershed state, including the areal mean tension water storage of the upper ($W_u$), lower ($W_l$), and deep
($W_d$) soil layers of the watershed, the areal mean free water storage ($S_0$), the storage of the interflow
linear reservoir ($O_i$), the storage of the groundwater linear reservoir ($O_g$), and the storage of the three
reservoirs in the Nash unit hydrograph ($F_1$, $F_2$ and $F_3$). Then, the hydrological response of the
XAJRNN unit at each time step is carried out through a "step_function", in which all computations are
encapsulated. The network calls the "step_function" in a sequential manner. The function mainly includes
the following four sub-functions:
(1)Based on the three-layer soil moisture model and the tension water capacity curve, the runoff
generation from the permeable portion of the watershed is calculated. The corresponding equations are
A1 to A12, and the main parameters involved include $A_{imp}$, $K_c$, $c$, $b$, $W_{um}$, $W_{lm}$, and $W_{dm}$. Then,
the areal mean tension water storage of the upper ($W_u$), lower ($W_l$), and deep ($W_d$) soil layers of the
watershed is updated, which will serve as the initial values for the next period. This corresponds to
equations A13 to A20. The main outputs obtained in this sub-function are the runoff ($R$), actual
evapotranspiration ($E_t$), and net rainfall ($P_n$).
(2) The runoff ($R$) is divided into different components using the free water capacity curve. The
corresponding equations are A21 to A25, and the parameters involved include $S_m$, $ex$, $K_i$, and $K_g$.
Then, the areal mean free water storage ($S_0$) is updated according to equation A26. The outputs obtained
in this sub-function are surface runoff ($R_s$), interflow runoff ($R_i$), and groundwater runoff ($R_g$).
(3) The interflow runoff ($R_i$) and groundwater runoff ($R_g$) obtained earlier are routed over the
hillslope using the linear reservoir method. The corresponding equations are A27 and A28, with
parameters $c_i$ and $c_g$. Then, combined with the surface runoff ($R_s$), the total inflow ($Q_t$) into the
channel network is calculated, corresponding to equation A29. The outputs obtained are the outflow of
the interflow linear reservoir ($Q_i$), the outflow of the groundwater linear reservoir ($Q_g$), and total inflow
($Q_t$).
(4) The Nash unit hydrograph method is used to perform channel network routing, resulting in the
final outflow ($Q$). The corresponding equations are A30 to A33, and the parameter involved is $K_f$. All
of the above physical parameters are automatically adjusted during the model training process through
gradient descent and backpropagation.

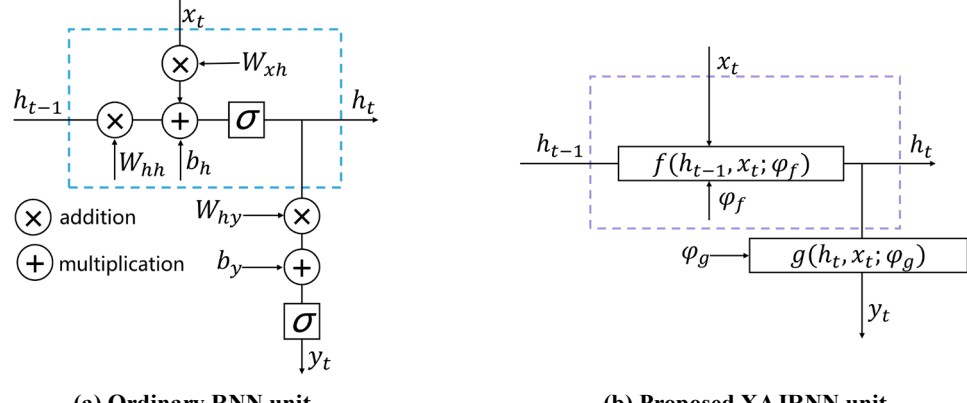

(a) Ordinary RNN unit                    (b) Proposed XAJRNN unit

**Figure 3: The structures of ordinary RNN unit and proposed XAJRNN unit.**
**3.2 Model setup**
**3.2.1 EDL model**
The proposed EDL model consists of the inputs, three neural network layers, and the outputs. First,
the XAJRNN layer, as discussed in Section 3.1.2, processes the input data to generate outputs. This neural
network layer follows the water balance principle and uses the physical sub-processes of the XAJ model
to describe the runoff generation and routing process. The output variables, significantly influenced by
the runoff process, are then passed to the Normalization layer. The purpose of this layer is to normalize
the data, helping the EDL model converge faster during training, increasing training stability, and
reducing the impact of differences between features. Specifically, the normalization layer adjusts the data
so that the mean is 0 and the standard deviation is 1. The normalized data is then passed into the LSTM
neural network layer for training. The choice of LSTM layers is based on two primary considerations:
first, its memory cells can retain hydrological information over extended periods, effectively capturing
the temporal dependencies of the rainfall–runoff process to enhance flood simulating accuracy; and
second, many studies have demonstrated that LSTM consistently improves hydrological model
simulation performance. For example, Alizadeh et al. (2021) demonstrated the SAINA-LSTM model
outperforms the EnsPost and MS-EnsPost in low, medium, and high flow ranges, as well as in 1 to 7 day
forecast horizons, and significantly reduces the root mean square error of flow predictions. Additionally,
Xu et al. (2022) combined the particle swarm optimization (PSO) algorithm with the LSTM model to
obtain the PSO-LSTM model. The research results show that the PSO-LSTM model outperforms the
Artificial Neural Network (ANN) and PSO-ANN at all stations in the basin. Finally, the trained EDL
model outputs the simulated runoff.

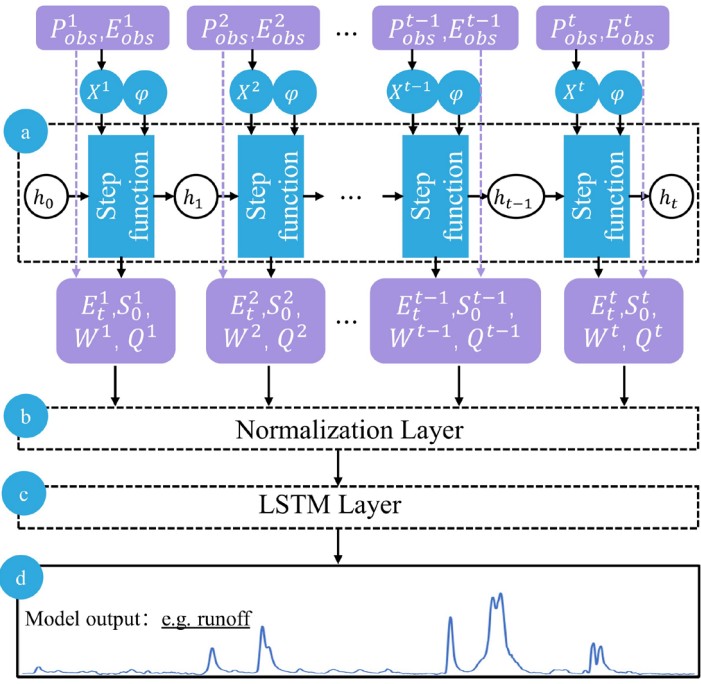

**Figure 4: The structure of the proposed EDL model. (a) The structure network schematic graph of the**
**XAJRNN layer. _h_ represents the state variables in the XAJRNN layer (as shown in Table A1). _φ_ represents**
**the parameters in the XAJRNN layer (as shown in Table A3). (b), (c), and (d) represent the normalization**
**layer, LSTM layer, and output results, respectively.**
For the EDL model, similar to the traditional XAJ model, the XAJRNN layer takes areal mean
rainfall ($P_{obs}$) and pan evaporation ($E_{obs}$) as input data, with a shape of [batch size, sequence length, 2
(input feature dimensions)]. The output physical quantities of interest are the actual evapotranspiration
($E_t$), the areal mean free water storage ($S_0$), the areal mean tension water storage ($W$), and outflow
discharge of the basin ($Q$). The selection of these four variables as the output of the XAJRNN layer is
primarily based on their high hydrological relevance to flood forecasting. Actual evapotranspiration ($E_t$)
is a key component of the hydrological cycle, directly affecting water availability and playing a crucial
role in runoff processes and flood simulation. Areal mean free water storage ($S_0$) and tension water
storage ($W$) represent the states of free water and water under tension in the watershed, reflecting the
basin storage capacity, which in turn influences flood occurrence and intensity. Outflow discharge ($Q$),
as the direct output of the basin system, is a core indicator for flood simulation. The selection of these
variables fully considers their physical significance and practical application value in flood simulation.
These four physical quantities, along with the two input sequences, form a new sequence that serves as
input for subsequent layers. The shape of the new input is [batch size, sequence length, 6 (new input
feature dimensions)]. After passing through normalization layers and LSTM layers, the final simulated
flow sequence is obtained. Following the general optimization methods of DL models, the parameters of
the XAJRANN and LSTM layer in the EDL model are optimized using gradient descent, specifically
with the *Adam* optimizer, to minimize the loss function. In our study, a separate model was trained for
each basin for flood simulation and forecasting, whose parameters were directly updated using the
standard end-to-end backpropagation approach. The model is trained with *NSE* as the loss function and
a learning rate of 0.001. The maximum number of iterations is set to 200, and training samples are reused
in each training cycle until convergence is achieved (i.e., the absolute difference in *NSE* between
consecutive cycles is less than 0.001).
**3.2.1 Benchmark model**
To compare the performance of the EDL model, three benchmark models are established. The first
benchmark model is the ordinary XAJ model, which also takes areal mean rainfall and evaporation as
input to illustrate the role of the XAJRNN layer in the EDL model. The input is the observed areal mean
rainfall ($P_{obs}$) and pan evaporation ($E_{obs}$), the output is the simulated flow discharge. Unlike the previous
DL model, we use the genetic algorithm (GA) to calibrate model parameters. The GA searches in a
population of points, uses the encoding of parameter sets, and uses probabilistic transition rules. There
are four GA hyperparameters: crossover probability parameter ($p_c$), mutation probability parameter ($p_m$),
population size parameter ($p_{size}$) and the maximum number of generation ($T_{max}$). Referring to the research
results of Cheng et al. (2006), the above hyperparameters are set to, $p_c$=0.8, $p_m$=0.1, $p_{size}$=150, and
$T_{max}$=1500. The second benchmark model is LSTM model, which differs from the EDL model only in
the absence of the XAJRNN layer; the rest of the architecture, including the normalization layer and
LSTM layer, remains the same. The purpose of LSTM model is to compare the contribution of XAJRNN

layer to the simulation performance in the EDL model. In order to reduce the impact of the training

process on the model performance, the training process and hyperparameters of the LSTM model are the

same as those of the EDL model.

**Table 1: Optimal parameters of the ordinary XAJ model calibrated using the GA algorithm.**

| Parameter | Value range | Lushui River basin | Qingjiang River basin |
|---|---|---|---|
| $K_c$ | [0.6,1.5] | 0.95 | 0.85 |
| $c$ | [0.01,0.2] | 0.18 | 0.19 |
| $W_{um}$ | [5,30] | 28.75 | 23.15 |
| $W_{lm}$ | [60,90] | 84.36 | 64.47 |
| $W_{dm}$ | [15,60] | 23.19 | 15.60 |
| $A_{imp}$ | [0.01,0.2] | 0.02 | 0.01 |
| $b$ | [0.1,0.4] | 0.40 | 0.35 |
| $S_m$ | [10,50] | 49.97 | 39.86 |
| $ex$ | [1,1.5] | 1.08 | 1.06 |
| $K_i$ | [0.1,0.55] | 0.19 | 0.37 |
| $K_g$ | [0.7-$K_i$] | 0.51 | 0.33 |
| $c_i$ | [0.1,0.9] | 0.87 | 0.89 |
| $c_g$ | [0.9,0.988] | 0.98 | 0.97 |
| $K_f$ | [0.1,10] | 3.99 | 1.58 |

The third benchmark model is the XAJ-LSTM hybrid model, which utilizes the simulated discharge

generated by the ordinary XAJ model as its primary input, augmented by observed areal mean rainfall

and pan evaporation data. The final output of XAJ-LSTM hybrid is the simulated flow discharge.

Similarly, the training process and hyperparameter configurations for the XAJ-LSTM model are kept

consistent with those used in the two previous benchmark models. The purpose of this benchmark model

is to demonstrate the superior performance of the proposed EDL model in comparison to using the LSTM

layers solely for hydrological post-processing.

**3.3 Evaluation metrics**

The overall performance of the models is evaluated using *NSE* (Nash and Sutcliffe, 1970), relative

error (*RE*), and root mean squared error (*RMSE*). The calculation formulas are as follows:

$$NSE = 1 - \frac{\sum_{i=1}^{N}(Q_{o,i}-Q_{f,i})^2}{\sum_{i=1}^{N}(Q_{o,i}-\bar{Q}_o)^2} \tag{4}$$

$$RE = \frac{\sum_{i=1}^{N} Q_{f,i} - \sum_{i=1}^{N} Q_{o,i}}{\sum_{i=1}^{N} Q_{o,i}} \times 100\% \tag{5}$$

$$RMSE = \sqrt{\frac{\sum_{i=1}^{N}(Q_{f,i} - Q_{o,i})^2}{N}} \tag{6}$$

where $N$ is the number of samples, $Q_o$, $\bar{Q}_o$ and $Q_{f,i}$ represent the observed inflows, mean value,
and simulated inflows, respectively.
To further evaluate the performance of the four models for flood event simulation, the flood peak
relative error ($PRE$) and the flood peak timing difference ($\Delta T$) are calculated by the following formulas:
$$PRE = \frac{Q_{f,peak} - Q_{o,peak}}{Q_{o,peak}} \times 100\% \tag{7}$$

$$\Delta T = T_o - T_f \tag{8}$$

where $Q_{o,peak}$ and $Q_{f,peak}$ represent the observed and simulated peak inflow discharge. $T_o$ and $T_f$
are the observed and simulated times of peak discharges occurred. If $\Delta T$ is positive, the simulated peak
discharge occurs early than the observed peak discharge; and vice versa.
**4 Results**
**4.1 Comparison of model performance**
Table 2 presents the evaluation metrics for flood simulation using four models (EDL, XAJ, LSTM,
and XAJ-LSTM) in the Lushui River and Qingjiang River basins. The evaluation metrics include *NSE*,
*RE*, and *RMSE* values for both training and test phases. In the Lushui River basin, the EDL model
demonstrated outstanding performance in both training and test periods. For the whole flow data series,
EDL achieved an *NSE* of 0.98 during the test period, with the lowest *RMSE* (43.71 m³/s) and a small
relative error (*RE* = -2.69%). These results outperformed both XAJ (*NSE* = 0.90, *RMSE* = 89.60 m³/s),
LSTM (*NSE* = 0.96, *RMSE* = 54.27 m³/s), and XAJ-LSTM (*NSE* = 0.92, *RMSE* = 73.54 m³/s). A similar
trend was observed in the Qingjiang River basin. the EDL model achieved an *NSE* of 0.92 and *RMSE* of
167.94 m³/s, maintaining a lower error compared to XAJ (*RMSE* = 231.17 m³/s), LSTM (*RMSE* = 155.71
m³/s), and XAJ-LSTM (*RMSE* = 227.62 m³/s). These results indicate that the EDL model generalizes
well to different hydrological conditions.
Furthermore, as noted in Section 3.1.2, the XAJRNN layer within the EDL model can directly output

simulated outflow (Q). To evaluate its performance, we extracted the runoff from the XAJRNN layer and compared it against observed streamflow in these two basins. The results are described as follows: in the Lushui River basin, the training period yielded *NSE*=0.92, *RE*=4.02%, *RMSE*=74.69 m³/s, while the testing period yielded *NSE*=0.90, *RE*=10.87%, *RMSE*=86.98 m³/s. In the Qingjiang River basin, the training period achieved *NSE*=0.89, *RE*=3.84%, *RMSE*=172.64 m³/s, and the testing period *NSE*=0.86, *RE*=-7.17%, *RMSE*=198.74 m³/s. Compared with the XAJ model, the runoff simulated by the XAJRNN layer shows overall improvement. However, its accuracy still falls short of the full EDL model. These findings confirm that while the XAJRNN layer has advantages over the standard XAJ model, integrating it with the LSTM layer could improve simulation accuracy.

It is important to note that the *RMSE* values of the two basins in Table 2 differ significantly. Specifically, the *RMSE* in the Lushui River basin is noticeably lower than that in the Qingjiang River basin. A possible reason for this difference is that, based on statistical calculations, the annual average flow of the Qingjiang River basin is 290 m³/s, whereas that of the Lushui River basin is only 96 m³/s. Additionally, the overall simulation performance in the Lushui River basin is better than in the Qingjiang River basin. During the test phase, the LSTM model demonstrated better simulation performance in the Qingjiang River basin, primarily due to the close integration of our EDL model with the XAJ model. Specifically, the XAJRNN layer in the EDL model adopts the runoff generation and routing principles of the XAJ model, and the model performance is closely related to the simulation accuracy of the XAJ model. When the XAJ model performs well, the EDL model also achieves better *NSE*, *RE*, and *RMSE* for the Qingjiang River.

**Table 2: Comparative analysis of model simulation accuracy evaluation metrics.**

| Basin | Model | Training period | | | Test period | | |
|---|---|---|---|---|---|---|---|
| | | *NSE* | *RE* (%) | *RMSE* (m³/s) | *NSE* | *RE* (%) | *RMSE* (m³/s) |
| Lushui River | EDL | 0.98 | 1.59 | 34.11 | 0.98 | -2.69 | 43.71 |
| | XAJ | 0.86 | -26.07 | 93.83 | 0.90 | -18.50 | 89.60 |
| | LSTM | 0.97 | -1.90 | 44.87 | 0.96 | -0.61 | 54.27 |
| | XAJ-LSTM | 0.93 | 4.24 | 70.90 | 0.92 | 19.06 | 73.54 |
| Qingjiang River | EDL | 0.95 | 1.10 | 104.09 | 0.92 | -8.74 | 167.94 |
| | XAJ | 0.85 | 5.91 | 182.05 | 0.85 | -7.92 | 231.17 |
| | LSTM | 0.90 | -4.16 | 147.89 | 0.93 | -6.19 | 155.71 |
| | XAJ-LSTM | 0.88 | 1.56 | 164.52 | 0.86 | -11.80 | 227.62 |

Figure 5 and Figure 6 respectively present the scatter plots of flood simulation results for the Lushui

River and Qingjiang River basins using the EDL model, the XAJ model, the LSTM model, and XAJ-

LSTM model. In Figure 5(a), during the training period, the scatter points of the EDL model are tightly

clustered and evenly distributed around the 1:1 ideal line. However, during the test period, in the range

where observed flow exceeds 3,000 $m^3/s$, most scatter points are located below the 1:1 ideal line. As

shown in Figure 5(b), the scatter points of the XAJ model are generally more dispersed compared to the

EDL model, and fall noticeably below the 1:1 ideal line across both the training and test periods. In

Figure 5(c), the scatter points of the LSTM model during the training period are evenly distributed around

the 1:1 ideal line but are more dispersed than those of the EDL model. In Figure 5(d), the scatter points

of the XAJ-LSTM model exhibit a distribution that lies between those of the XAJ and LSTM models,

with a higher degree of dispersion compared to the tightly grouped points of the EDL model. During the

test period, the scatter points in the low to medium flow range are evenly distributed around the 1:1 ideal

line, similar to the EDL model, but in the range where the observed flow exceeds 3,000 $m^3/s$, most scatter

points are below the 1:1 ideal line. The reason that the scatter points fall below the 1:1 ideal line in the

range where the observed flow exceeds 3,000 $m^3/s$ may be due to the fact that during the training period,

there were few flow values exceeding 3,000 $m^3/s$, while in the test period, there were relatively more

high flows exceeding 3,000 $m^3/s$. In summary, it can be concluded that the scatter plots of the EDL model

are relatively better, while the scatter plots of the XAJ and LSTM models are relatively worse.

In Figure 6 (a), the scatter points of the EDL model are very compact and evenly distributed on both

sides of the 1:1 ideal line during the training period. However, the scatter points are more loosely

distributed, and some scatter points are obviously below the 1:1 ideal line during the test period. As

shown in Figure 6 (b), the scatter points of the XAJ model during the training period are more scattered

than the EDL model, and the scatter points are obviously deviated from the 1:1 ideal line in the range

where observed flow exceeds 4,000 $m^3/s$. During the test period, the scatter point distribution of the XAJ

model is looser, and the scatter points are farther from the 1:1 ideal line in the range where the observed

flow exceeds 4,000 $m^3/s$. As shown in Figure 6 (c), the scatter point distribution of the LSTM model

during the training period is similar to that of the EDL model. During the test period, it is also similar to

the EDL model, but the scatter points are obviously below the 1:1 ideal line in the range where the
observed flow exceeds 6,000 m³/s. In Figure 6(d), the scatter points of the XAJ-LSTM model are more
dispersed around the 1:1 ideal line than those of the EDL model. In summary, it can be concluded that
the scatter plots of the EDL model are relatively better than these of the XAJ and LSTM models.

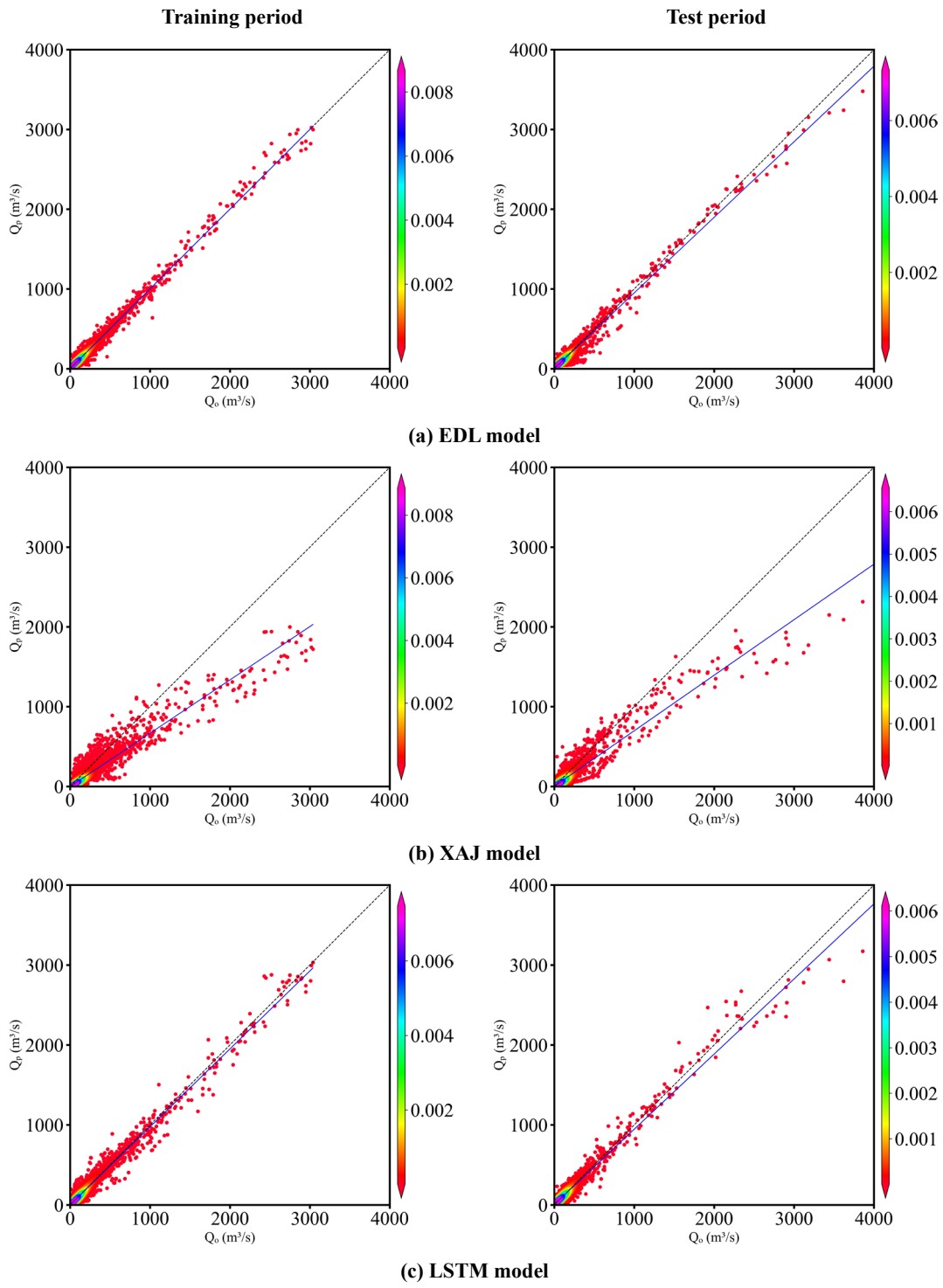

**(a) EDL model**

**(b) XAJ model**

**(c) LSTM model**

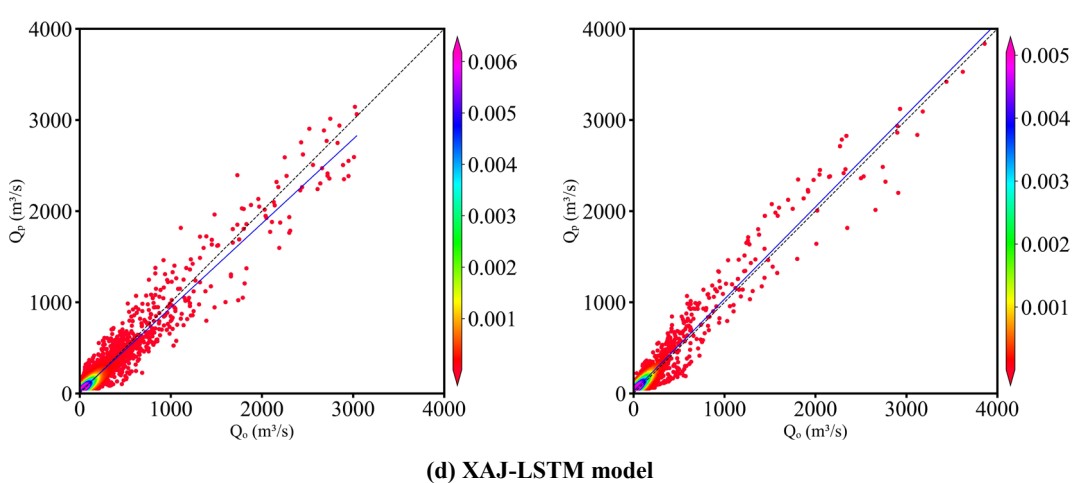

**(d) XAJ-LSTM model**

Figure 5: Scatter plots of observed ($Q_o$) and simulated ($Q_p$) flow discharges by four models in the Lushui River basin. The color bar represents the density of the scatter distribution. The denser the scatter distribution, the higher the corresponding density value in color.

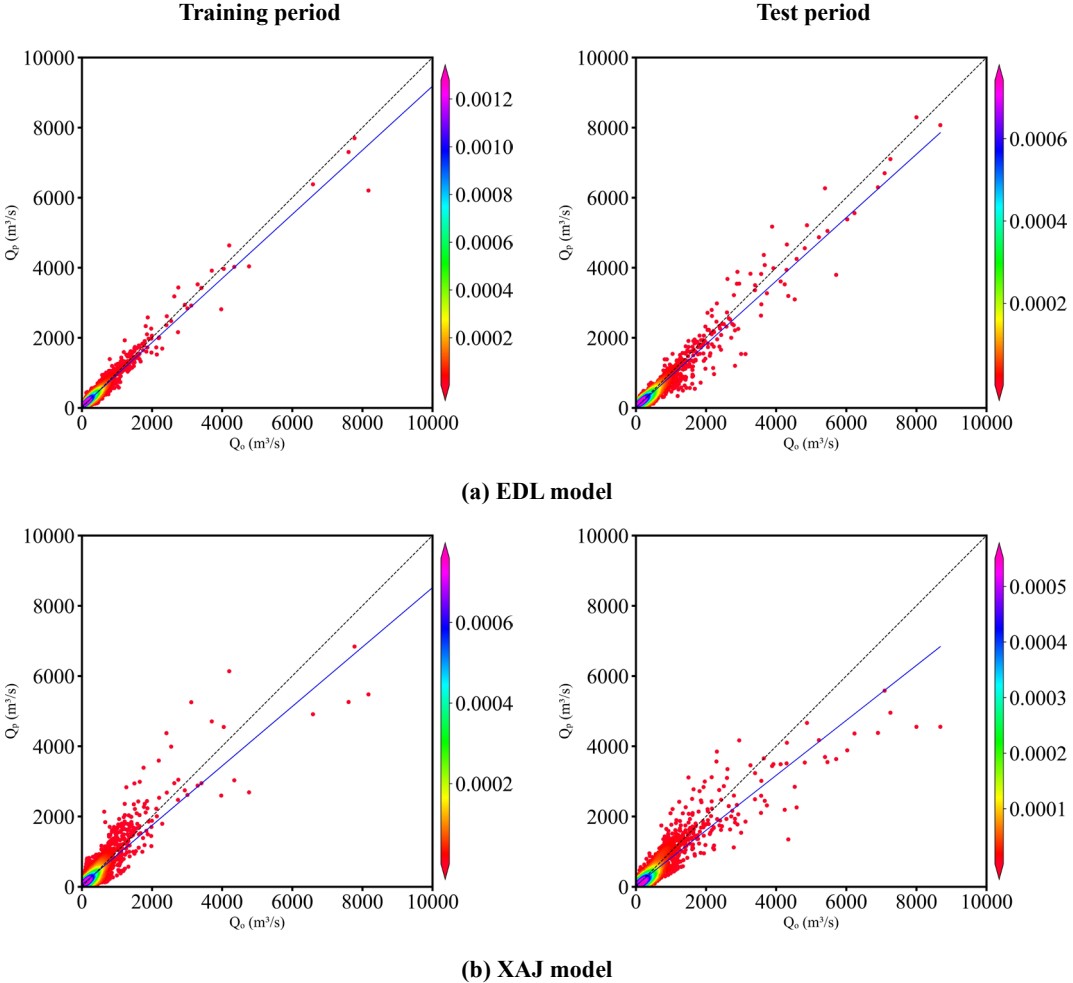

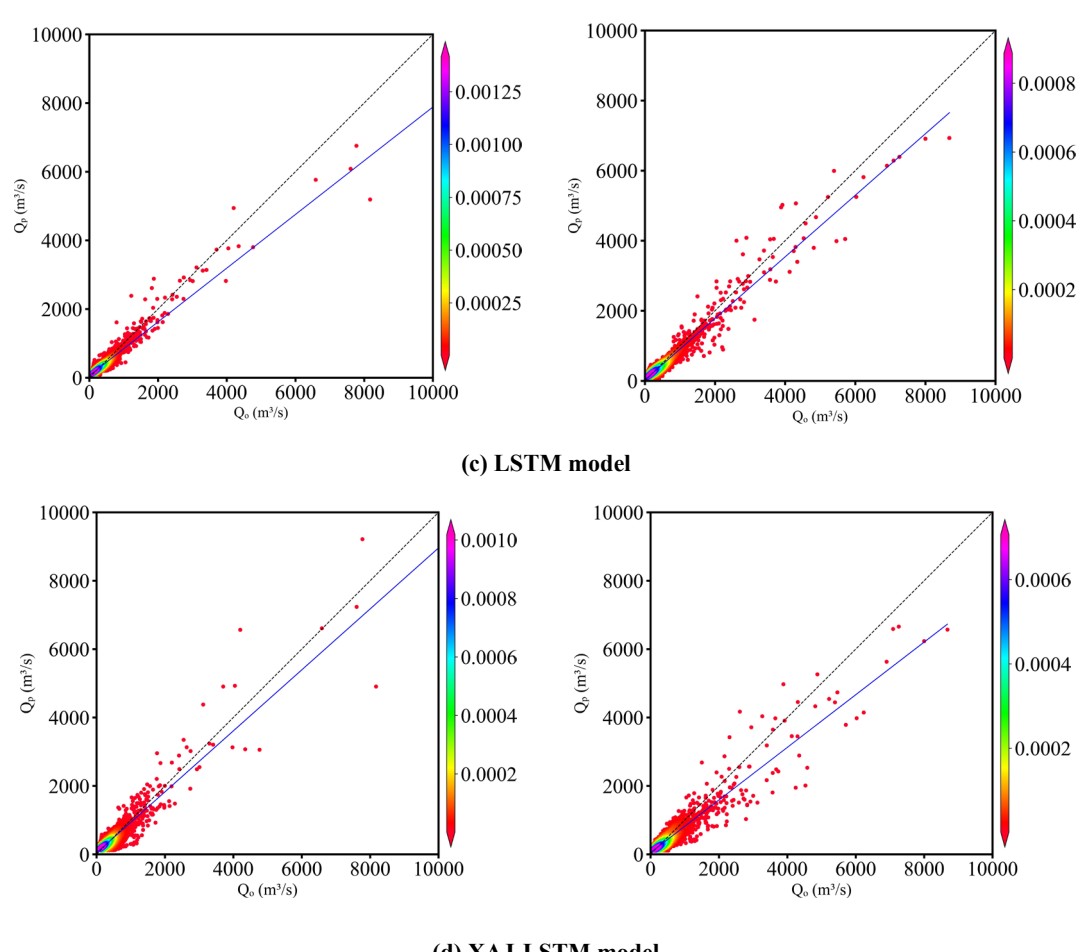

**(c) LSTM model**

**(d) XAJ-LSTM model**

Figure 6: Scatter plots of observed ($Q_o$) and simulated ($Q_p$) flow discharge by four models in the Qingjiang River basin. The color bar represents the density of the scatter distribution. The denser the scatter distribution, the higher the corresponding density value in color.

**4.2 Comparison of the effectiveness of flood events simulation**

Four major flood processes during the test period were selected in the Lushui River and Qingjiang River basins as case study. The simulation evaluation metrics of the three models (EDL, XAJ, LSTM, and XAJ-LSTM) in these four flood processes are shown in Tables 3 and 4, respectively. These evaluation metrics include *NSE*, *RE*, *RMSE*, *PRE*, and $\Delta T$. It should be specifically noted that if $\Delta T$ is positive, the simulated peak discharge occurs earlier than the observed peak discharge; and vice versa.

As shown in Table 3, the EDL model performed exceptionally well with high simulation accuracy, the *NSE* ranged from 0.97 to 0.99, *RE* ranged from -7.54% to -1.9%, and *RMSE* was as low as 71.66 m³/s. Additionally, the EDL model's *PRE* was consistently below -12%, and $\Delta T$ remained at 0, highlighting its high reliability in simulating peak flow magnitude and timing. In contrast, the XAJ

model's *NSE* ranged from 0.65 to 0.93, with significant *RE* deviations and *RMSE* values much higher than those of the EDL model, resulting in subpar overall performance. The LSTM model's *NSE* ranged from 0.91 to 0.97, close to that of the EDL model, but its *RE* and *RMSE* were slightly less favorable, resulting in marginally lower simulation accuracy. The XAJ-LSTM model achieves a slightly lower NSE value compared to the EDL model, along with considerably higher RE and RMSE, indicating an overall inferior predictive performance.

As shown in Table 4, the EDL model continued to demonstrate superior performance, with *NSE* exceeding 0.95 in all cases except for extreme events, *RE* ranging from -4.7% to -0.17% with minimal bias, and *RMSE* as low as 266.62 m³/s. Although the EDL model's performance slightly declined during extreme events (e.g., 20200726), it still outperformed other models overall. The XAJ model's performance in the Qingjiang River basin was significantly inferior to the EDL model, with *NSE* varying widely, reaching as low as 0.32, *RE* deviations as high as 26.42%, and *RMSE* peaking at 1277.61 m³/s, indicating its poor adaptability to complex events. The LSTM model's *NSE* ranged from 0.64 to 0.94, overall better than the XAJ model, but its accuracy and timeliness in peak flow simulation were insufficient during extreme events. For the 20200628-flood event, the performance of the XAJ-LSTM model was comparable to that of the EDL model. While for the other three flood events, the EDL-based approach performed much better than the XAJ-LSTM model.

**Table 3: Flood simulation evaluation metrics for different events in the Lushui River basin.**

| Flood event | Model | *NSE* | *RE* (%) | *RMSE*(m³/s) | *PRE*(%) | $\Delta T$(h) |
|---|---|---|---|---|---|---|
| 20170624 | EDL | 0.98 | -7.54 | 211.98 | -11.99 | 0 |
| | XAJ | 0.83 | -19.31 | 561.31 | -26.62 | 0 |
| | LSTM | 0.91 | -12.29 | 421.53 | -21.7 | -3 |
| | XAJ-LSTM | 0.91 | -6.79 | 417.77 | -20.84 | -3 |
| 20170702 | EDL | 0.97 | -3.53 | 138.01 | -11.57 | 0 |
| | XAJ | 0.65 | -24.8 | 468.94 | -37.34 | 30 |
| | LSTM | 0.93 | 0.76 | 204.43 | -12.59 | 27 |
| | XAJ-LSTM | 0.83 | -6.28 | 325.57 | -18.16 | 30 |
| 20170813 | EDL | 0.99 | -1.9 | 71.66 | -0.9 | 0 |
| | XAJ | 0.85 | -18.28 | 351.25 | -26.61 | 3 |
| | LSTM | 0.97 | 0.28 | 142.23 | -7.31 | 0 |
| | XAJ-LSTM | 0.92 | 2.67 | 255.68 | -1.86 | 3 |
| 20190526 | EDL | 0.98 | -3.85 | 85.92 | -0.78 | 0 |
| | XAJ | 0.93 | 1.23 | 175.86 | -10.15 | 0 |
| | LSTM | 0.97 | -4.49 | 109.74 | 0.45 | 0 |
| | XAJ-LSTM | 0.97 | -0.76 | 113.95 | -5.66 | 0 |

**Table 4: Flood simulation evaluation metrics for different events in the Qingjiang River basin.**

| Flood event | Model | NSE | RE (%) | RMSE(m³/s) | PRE(%) | ΔT(h) |
|---|---|---|---|---|---|---|
| 20171003 | EDL | 0.95 | -4.43 | 244.26 | -7.64 | 0 |
| | XAJ | 0.89 | 5.38 | 357.55 | -8.59 | 0 |
| | LSTM | 0.85 | -15.82 | 417.92 | -27.02 | 0 |
| | XAJ-LSTM | 0.87 | -9.36 | 382.81 | -13.37 | 0 |
| 20200628 | EDL | 0.98 | -0.17 | 266.62 | -2.2 | 0 |
| | XAJ | 0.95 | 5.48 | 467.13 | -7.34 | 0 |
| | LSTM | 0.94 | -3.58 | 506.72 | -11.95 | 0 |
| | XAJ-LSTM | 0.97 | -2.80 | 377.35 | -8.29 | 0 |
| 20200717 | EDL | 0.95 | -1.74 | 533.14 | -4.48 | 6 |
| | XAJ | 0.66 | -23.15 | 1368.23 | -28.61 | 0 |
| | LSTM | 0.91 | -5.43 | 696.92 | -20.18 | 0 |
| | XAJ-LSTM | 0.78 | -19.29 | 1100.59 | -24.34 | 0 |
| 20200726 | EDL | 0.61 | -4.7 | 966.73 | -9.45 | -6 |
| | XAJ | 0.32 | 26.42 | 1277.61 | -12.75 | -6 |
| | LSTM | 0.64 | 1.84 | 931.87 | -13.29 | -6 |
| | XAJ-LSTM | 0.51 | 8.13 | 1089.91 | -12.94 | -6 |

In summary, the EDL model exhibited the best overall performance in flood simulations for both the Lushui and Qingjiang River basins, with high accuracy, low bias, and excellent stability, particularly in regular flood events. Although the performance of LSTM and XAJ-LSTM models were close to that of the EDL model overall, it was slightly lacking in extreme events. In comparison, the XAJ model lagged significantly in both accuracy and adaptability, making it less suitable for precise flood simulation.

The outstanding performance of the EDL model highlights its immense potential in flood simulation, especially in complex basin conditions and extreme flood events. This further proves the advancement and feasibility of the model obtained by coupling DL technology with traditional hydrological models in the field of hydrological simulation, and provides strong tool support for solving flood forecasting problems.

To visually demonstrate the advantages of the EDL model, Figure 7 and 8 respectively present the hydrographs of four flood events in the Lushui River and Qingjiang River basins, comparing the simulation results of the EDL model, XAJ model, LSTM model, and XAJ-LSTM model.

From Figure 7, it can be observed that the 20170624-flood event, all four models underestimated the peak flow discharge to varying degrees, but the EDL model performed relatively better and more accurately simulated the timing of the peak flow. In the 20170702 and 20170813 compound flood events,

the EDL model's simulated hydrograph during the peak flow was closer to the observed hydrograph
compared to the three benchmark models. For 20190526-flood event, both the EDL model and the LSTM
model simulated the peak flow discharge well. However, in the 20170702 and 20190526 flood events,
all four models exhibited delays, as evidenced by discrepancies in the rising speed during the flood rising
phase compared to the observations. This may be due to the slow response of the model to rainfall.
Overall, the EDL model performed well in simulating the hydrographs of the Lushui River basin,
accurately capturing both the peak flow discharge and the timing of the peak.

Compared to the Lushui River basin, the simulation results of the four models in the Qingjiang

River basin showed certain limitations, which were particularly evident in the 20200726-flood event as
shown in Figure 8 (d). The poor simulation performance may be attributed to two major influencing
factors. First, the location of the heavy rainfall center has a significant impact on the simulation results.
Since the model input uses areal average rainfall, it fails to fully account for the spatial distribution
characteristics of rainfall. As shown in Figure 8(d), when the heavy rainfall center is close to the Shuibuya
Reservoir, the short flow routing time leads to a significant decline in the model simulation performance.
Second, the impact of upstream reservoir regulation cannot be ignored. During multiple flood events in
the Qingjiang River basin in 2020, the Shuibuya Reservoir increased outflow discharge to cope with
severe flood control conditions.

All four models underestimated the peak flow discharge, and the simulated peak flow time was

significantly delayed compared to the observed flow peak time. Our study focused on two basins: the
Lushui and Qingjiang River basins. As illustrated in Figure 1, the Qingjiang River basin features a more
complex terrain and a more meandering river network compared to the Lushui River basin. Based on the
flow simulation results shown in Figures 7 and 8, the model performance in the Lushui River basin is
better than that in the Qingjiang River basin. These findings suggest that the model simulation accuracy
in simple terrain basin is higher than that in the complex terrain conditions. For 20201003 and 20200717
flood events, the EDL model's simulated hydrograph was closer to the observed hydrograph compared
to the benchmark models. For 20200628-flood event, the LSTM model performed better during the
recession phase but significantly underestimated the peak flow discharge and failed to accurately
simulate the rising phase. In contrast, the EDL model performed better during the rising and peak phases
but exhibited delays during the recession phase. Overall, although some deviations in peak flow
discharge and timing exist, the EDL model still effectively captures the general flood trends in the
Qingjiang River basin.

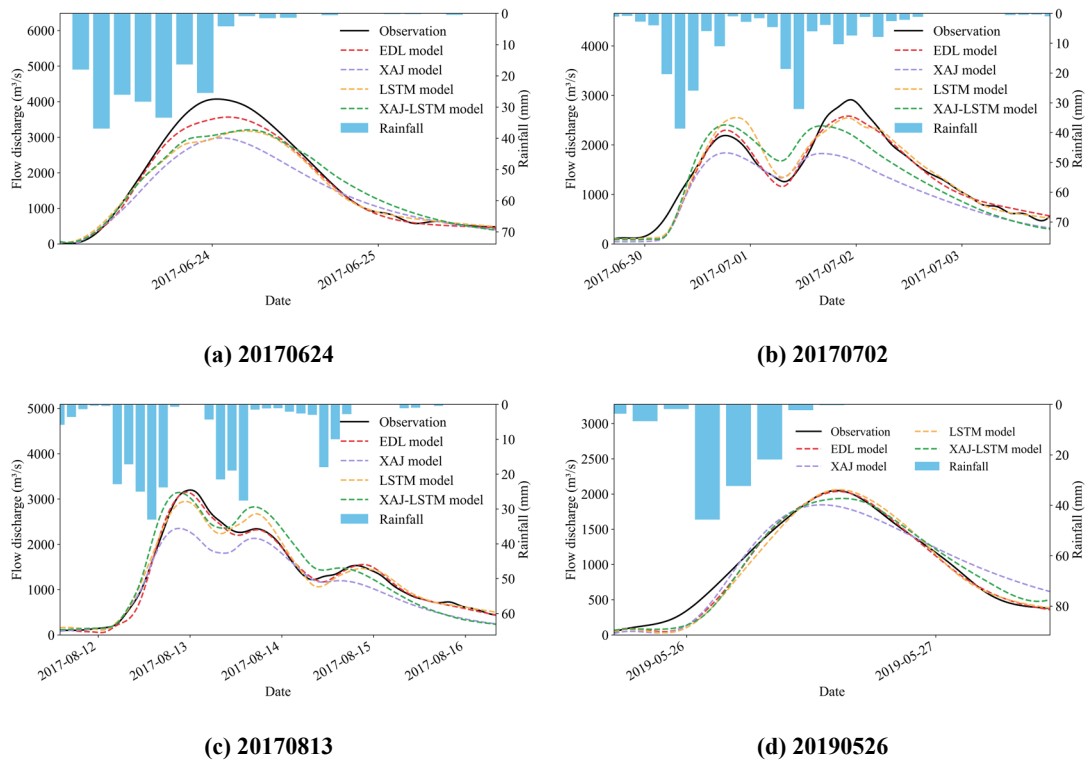

(a) 20170624                    (b) 20170702

(c) 20170813                    (d) 20190526

**Figure 7: Comparison of observed and simulated flood hydrographs in the Lushui River basin.**

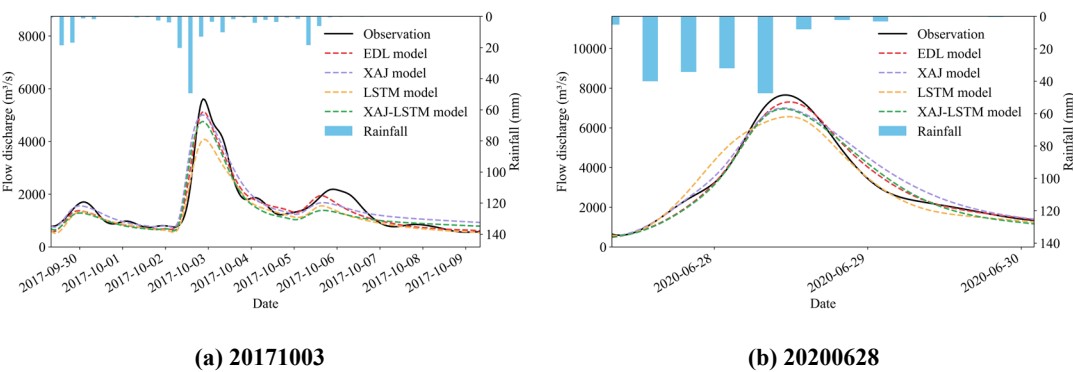

(a) 20171003                    (b) 20200628

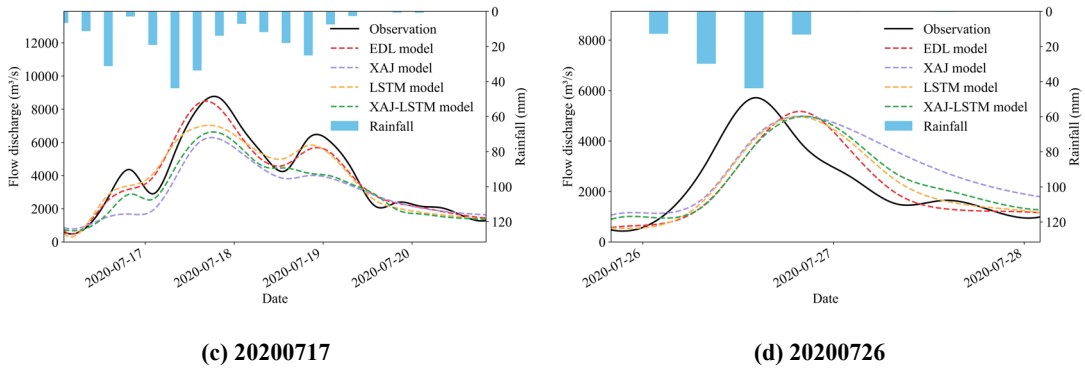

<p align="center"><b>(c) 20200717</b>        <b>(d) 20200726</b></p>

**Figure 8: Comparison of observed and simulated flood hydrographs in the Qingjiang River basin.**
**5 Discussion**

The XAJ model is a well-established hydrological model utilized for generalizing hydrological

processes in a basin, including runoff generation and routing. However, when used in isolation, the model
may struggle to adequately capture intricate nonlinear relationships, particularly evident in flood peak
simulating where it might not fully account for the impacts of real-time meteorological changes. On the
other hand, DL models, especially time-series models like LSTM, are adept at capturing complex
nonlinear relationships within time series data. Nevertheless, they may encounter delays in accurately
simulating flood peak timings (Chen et al., 2022; Cui et al., 2021; Xiang et al., 2024). To address this
limitation, a fusion of the XAJ model and DL models can mitigate the weaknesses inherent in each
approach. Specifically, the conventional hydrological model offers a foundation for physical processes
that enhance the simulation of basin hydrological responses, while the DL model can refine the output
of the hydrological model, particularly in terms of temporal accuracy and the comprehension of nonlinear
relationships. This hybrid approach allows the XAJ model to capture long-term dependencies in
hydrological processes while enabling the DL model to make more accurate simulations regarding flood
peak timing, thus effectively minimizing delays in flood peak time.

In traditional processes, DL models such as LSTM are commonly employed as post-processors to

correct the outputs of hydrological models (Cho and Kim, 2022; Cui et al., 2021; Frame et al., 2021; Han
and Morrison, 2022). Nevertheless, a significant drawback of this methodology stems from the
inconsistent parameter optimization. Typically, the parameters of the hydrological model are initially
calibrated to achieve the best simulation results, followed by the training of an LSTM model for

<p align="center">27</p>

refinement. Since the parameter adjustments of the hydrological and DL models are conducted
independently, the resulting parameter combination is often suboptimal, thereby constraining simulation
accuracy. A comparison between the EDL model and the XAJ-LSTM model highlights this issue. The
XAJ-LSTM model, which uses the outputs of the traditional XAJ model as inputs and is trained
independently, shows some improvement over the XAJ model but still underperforms compared with the
EDL model. By contrast, the proposed EDL model integrates hydrological and DL components within a
unified framework, enabling synchronized training and joint parameter optimization. This online strategy
not only eliminates the parameter mismatch inherent in conventional post-processing methods but also
ensures that both hydrological and DL parameters are optimized simultaneously, leading to generate
synergistic benefits.
This study demonstrates that the proposed EDL model, which tightly integrates physical
mechanisms with DL, can effectively improve the accuracy of flood simulation and forecasting. A
limitation of the present work is that the model parameters were obtained separately for each basin.
Consequently, the current implementation represents a locally trained model that is functionally similar
to traditional calibration. Future research should therefore pursue two complementary directions to
improve generality and adaptability. One is to develop multi-basin (regional or global) training strategies
via differentiable parameter learning, transfer learning, or meta-learning, which can leverage data from
many basins to produce a single, generalizable model. Another is to introduce mechanisms for dynamic,
input-dependent parameter adaptation so that model parameters can evolve with temporal changes in
inputs. Additional promising avenues include explicit uncertainty quantification, tighter coupling
between physics and data-driven components, and online-updating for real-time forecasting. Pursuing
these directions would increase the model applicability across diverse basins and enhance its potential
for scientific discovery.
**6 Conclusions**
The present study proposes a novel EDL model that combines the physics-driven XAJRNN layer
with the LSTM layer, successfully achieving accurate simulation of flood processes in the Lushui River
basin and Qingjiang River basin. This model leverages the relatively complex physical mechanisms of
the XAJ model and the nonlinear representation capabilities of DL, demonstrating strong simulation
performance. The key findings of this study are as summarized as follows:

(1) The EDL model demonstrates superior performance in both simulation accuracy and error

control. It achieves an average *NSE* of 0.98 in the Lushui River basin and 0.94 in the Qingjiang River
basin, demonstrating its outstanding fitting capability. Its average *RMSE* is 38.91 m³/s in the Lushui River
basin and 136.02 m³/s in the Qingjiang River basin, significantly lower than that of benchmark models,
highlighting its superior simulation accuracy. Although the *RE* is slightly higher during the testing phase,
the combined analysis of the training phase *RE* shows that the EDL model consistently outperforms its
counterparts with stable error control.

(2) The EDL model demonstrates the highest stability in most flood simulations. Compared to the

benchmark models, the EDL model achieves smaller *PRE* values, indicating its superior accuracy in
simulating flood peak magnitudes. Moreover, except for a few rare cases, the EDL model's $\Delta T$ is nearly
zero, showcasing its unparalleled precision in simulating the timing of flood peaks.

(3) Compared to traditional single models, the EDL model not only significantly improves

simulation accuracy but also enhances interpretability by integrating physical mechanisms. This
innovative approach paves the way for the seamless integration of DL with hydrological physical
mechanisms, advancing research in the field.
**Code availability**
The code used to support the findings of this study are available from the corresponding author upon
request.
**Data availability**
The data generated and/or analyzed during the current study are not publicly available for legal/ethical
reasons but are available from the corresponding author on reasonable request.

## Author contributions

X.X. and S.G. conceived and designed the experiments; X.X. performed the experiments and wrote the manuscript draft; X.X., S.G., C.L., and Y.W. reviewed and edited the manuscript.

## Competing interests

The authors declare that they have no conflict of interest.

## Acknowledgments

This study was financially supported by the National Natural Science Foundation of China (No. U2340205). The authors would like to thank the editor and anonymous reviewers whose comments and suggestions help to improve the manuscript.

**Appendix A. Supplementary tables and texts**
Table A1 describes the state variables of the XAJ model.
Table A2 describes the flux of the XAJ model.
Table A3 describes parameters and their value ranges of the XAJ model.
Table A4 describes the pseudocode of the XAJRNN layer.
Text A1 describes the details of the XAJ model.

**Table A1 State variables of the XAJ model.**

| Module | State variable | Meaning | Unit |
|---|---|---|---|
| Evapotranspiration | $W_u$ | Areal mean tension water storage of the upper soil layer | |
| | $W_l$ | Areal mean tension water storage of the lower soil layer | |
| | $W_d$ | Areal mean tension water storage of the deep soil layer | |
| Runoff separation | $S_0$ | Areal mean free water storage | |
| Flow routing | $O_i$ | Water storage of the interflow linear reservoir | |
| | $O_g$ | Water storage of the groundwater linear reservoir | mm |
| | $F_1$ | Water storage of the first reservoir in the Nash unit hydrograph | |
| | $F_2$ | Water storage of the second reservoir in the Nash unit hydrograph | |
| | $F_3$ | Water storage of the third reservoir in the Nash unit hydrograph | |


**Table A2 Flux of the XAJ model.**

| Module | Flux | Meaning | Unit |
|---|---|---|---|
| Evapotranspiration | $P_{obs}$ | Areal mean rainfall | $mm/\Delta t$ |
| | $E_{obs}$ | Measured pan evaporation | |
| | $P$ | Areal mean rainfall of the impervious area | |
| | $R_{imp}$ | Runoff directly from the impervious area | |
| | $P_n$ | Areal mean net rainfall | |
| | $E_p$ | Potential evapotranspiration | |
| | $E_u$ | Actual evapotranspiration of the upper soil layer | |
| | $E_l$ | Actual evapotranspiration of the lower soil layer | |
| | $E_d$ | Actual evapotranspiration of the deep soil layer | |
| | $E_t$ | Actual evapotranspiration | |
| Runoff generation | $R$ | Runoff produced from the previous area | $mm/\Delta t$ |
| Runoff separation | $R_s$ | Surface runoff | $mm/\Delta t$ |
| | $R_i$ | Interflow runoff | |
| | $R_g$ | Groundwater runoff | |
| Flow routing | $Q_i$ | Outflow of the interflow linear reservoir | $mm/\Delta t$ |
| | $Q_g$ | Outflow of the groundwater linear reservoir | |
| | $Q_t$ | Total inflow to channel network | |
| | $Q_1$ | Outflow of the first reservoir in the Nash unit hydrograph | |
| | $Q_2$ | Outflow of the second reservoir in the Nash unit hydrograph | |
| | $Q_3$ | Outflow of the third reservoir in the Nash unit hydrograph | |
| | $Q$ | Outflow discharge of the basin | $m^3/s$ |


**Table A3 Parameters and their value ranges of the XAJ model.**

| Module | Parameter | Meaning | Value range | Unit |
|---|---|---|---|---|
| Evapotranspiration | $K_c$ | Ratio of potential evapotranspiration to pan evapotranspiration | [0.6,1.5] | - |
| | $c$ | Coefficient of deep evapotranspiration | [0.01,0.2] | - |
| | $W_{um}$ | Areal mean tension water capacity of the upper soil layer | [5,30] | mm |
| | $W_{lm}$ | Areal mean tension water capacity of the lower soil layer | [60,90] | mm |
| | $W_{dm}$ | Areal mean tension water capacity of the deep soil layer | [15,60] | mm |
| | $A_{imp}$ | Ratio of the impervious area | [0.01,0.2] | - |
| Runoff generation | $b$ | Exponent of the tension water capacity curve | [0.1,0.4] | - |
| Runoff separation | $S_m$ | Areal mean of the free water capacity of the surface soil layer | [10,50] | mm |
| | $ex$ | Exponent of the free water capacity curve | [1,1.5] | - |
| | $K_i$ | Outflow coefficient of the free water storage to interflow | [0.1,0.55] | - |
| | $K_g$ | Outflow coefficient of the free water storage to groundwater | [0.7-$K_i$] | - |
| Flow routing | $c_i$ | Recession constant of interflow storage | [0.1,0.9] | - |
| | $c_g$ | Recession constant of groundwater storage | [0.9,0.988] | - |
| | $K_f$ | Storage-discharge coefficient of the linear reservoir in the Nash unit hydrograph | [0.1,10] | - |


**Table A4 Pseudocode of the XAJRNN unit.**

**Algorithm**: the XAJRNN unit

**Input**: Sequences of observed rainfall $\{P_{obs}\}$ and observed evapotranspiration $\{E_{obs}\}$

**State initialization**: $W_u^{(0)} = 0$, $W_l^{(0)} = 0$, $W_d^{(0)} = 0$, $S_0^{(0)} = 0$, $O_i^{(0)} = 0$, $O_g^{(0)} = 0$, $O_s^{(0)} = 0$, $F_1^{(0)} = 0$, $F_2^{(0)} = 0$ and $F_3^{(0)} = 0$

**Parameters**: $K_c$, $c$, $W_{um}$, $W_{lm}$, $W_{dm}$, $A_{imp}$, $b$, $S_m$, $ex$, $K_i$, $K_g$, $c_i$, $c_g$, $K_f$, $n$

**function** step_function ($[P^{(i)}, E^{(i)}]$, $[W_u^{(i-1)}, W_l^{(i-1)}, W_d^{(i-1)}, S_0^{(i-1)}, O_i^{(i-1)}, O_g^{(i-1)}, O_s^{(i-1)}, F_1^{(i-1)}, F_2^{(i-1)}, F_3^{(i-1)}]$, parameters):

    Calculate $R^{(i)}$, $R_{imp}^{(i)}$ $E_t^{(i)}$, $P_n^{(i)}$, $W_u^{(i)}$, $W_l^{(i)}$, $W_d^{(i)}$ via Eqs. (A1) – (A20)

    Calculate $R_s^{(i)}$, $R_i^{(i)}$, $R_g^{(i)}$, $S_0^{(i)}$ via Eqs. (A21) – (A26)

    Calculate $Q_i^{(i)}$, $Q_g^{(i)}$, $Q_t^{(i)}$, $O_i^{(i)}$, $O_g^{(i)}$ via Eqs. (A27) – (A29)

    Calculate $Q_1^{(i)}$, $Q_2^{(i)}$, $Q_3^{(i)}$, $F_1^{(i)}$, $F_2^{(i)}$, $F_3^{(i)}$ via Eqs. (A30) – (A32)

**return** $W_u^{(i)}$, $W_l^{(i)}$, $W_d^{(i)}$, $S_0^{(i)}$, $O_i^{(i)}$, $O_g^{(i)}$, $O_s^{(i)}$, $F_1^{(i)}$, $F_2^{(i)}$ and $F_3^{(i)}$

**do** RNN (step function, $[\{P\}, \{E\}]$, $[W_u^{(0)}, W_l^{(0)}, W_d^{(0)}, S_0^{(0)}, O_i^{(0)}, O_g^{(0)}, O_s^{(0)}, F_1^{(0)}, F_2^{(0)}, F_3^{(0)}]$)

to obtain sequences of $\{W_u\}$, $\{W_l\}$, $\{W_d\}$, $\{S_0\}$, $\{O_i\}$, $\{O_g\}$, $\{O_s\}$, $\{F_1\}$, $\{F_2\}$, $\{F_3\}$

Calculate sequence of $\{Q\}$ via Eq. (A33)

**Output**: The sequence of runoff at the catchment outlet $\{Q\}$


**Text A1**
In the evapotranspiration, considering the uneven vertical distribution of soil, the XAJ model
divides the soil into three layers and calculates the actual evapotranspiration using a three-layer soil
moisture model. The calculation principle is as follows: The upper layer evaporates according to its
evapotranspiration capacity. When the upper layer's water content is insufficient, the remaining
evapotranspiration capacity is supplied by evaporation from the lower layers. The evaporation from the
lower layers is proportional to the water storage in those layers. The ratio of the evaporation from the
lower layer to the remaining evapotranspiration capacity must not be less than the coefficient of deep
evapotranspiration ($c$). Otherwise, the lower layer water storage will supply the insufficient portion. If
the lower layer water storage is not sufficient to compensate, the deep layer water storage will provide
the remainder. The calculation formula is as follows:
$$P = P_{obs}(1 - A_{imp}) \tag{A1}$$

$$R_{imp} = P_{obs} \times A_{imp}$$

$$E_p = K_c E_{obs} \tag{A2}$$

(1) When $W_u + P \geq E_p$,
$$E_u = E_p; \ E_l = 0; \ E_d = 0 \tag{A2}$$

(2) When $W_u + P < E_p$ and $W_l \geq c \times W_{lm}$,
$$E_u = W_u + P; E_l = (E_p - E_u) \times W_l/W_{lm}; \ E_d = 0 \tag{A3}$$

(3) When $W_u + P < E_p$ and $c \times (E_p - E_u) \leq W_l < c \times W_{lm}$,
$$E_u = W_u + P; \ E_l = c \times (E_p - E_u); \ E_d = 0 \tag{A4}$$

(4) When $W_u + P < E_p$ and $c \times (E_p - E_u) > W_l$,
$$E_u = W_u + P; \ E_l = W_l; \ E_d = c \times (E_p - E_u) - E_l \tag{A5}$$

$$E_t = E_u + E_l + E_d \tag{A6}$$

$$P_n = \begin{cases} P - E_t, & P \geq E_t \\ 0, & P < E_t \end{cases} \tag{A7}$$

The runoff generation calculation uses the tension water capacity curve. First, it is necessary to
calculate the areal mean tension water storage ($W$) and the areal mean tension water capacity ($W_m$):
$$W = W_u + W_l + W_d \tag{A8}$$

$$W_m = W_{um} + W_{lm} + W_{dm} \tag{A9}$$

The vertical coordinate value ($\alpha$) corresponding to the areal mean tension water storage ($W$) on the
tension water capacity curve is calculated as:

$$\alpha = W_m \times (b + 1) \times \left[1 - \left(1 - \frac{W}{W_m}\right)^{\frac{1}{1+b}}\right] \qquad \text{(A11)}$$

Calculate the runoff produced from the previous area:

$$R = \begin{cases} P_n + W - W_m + W_m(1 - \frac{P_n + \alpha}{W_m \times (b+1)})^{b+1}, \ P_n + \alpha \leq W_m \times (b + 1) \\ P_n + W - W_m, \ P_n + \alpha > W_m \times (b + 1) \end{cases} \qquad \text{(A12)}$$

Finally, update the areal mean tension water storage of the upper, lower, and deep soil layer of the
watershed at the end of the current period, which will serve as the initial values for the next period:

$$W_u = W_u + P - E_t - R \qquad \text{(A13)}$$

$$W_l = W_l - E_l \qquad \text{(A14)}$$

$$W_d = max(W_d - E_d, 0) \qquad \text{(A15)}$$

When $W_u > W_{um}$,

$$W_l = W_l + W_u - W_{um} \qquad \text{(A16)}$$

$$W_u = W_{um} \qquad \text{(A17)}$$

When $W_l > W_{lm}$,

$$W_d = W_d + W_l - W_{lm} \qquad \text{(A18)}$$

$$W_l = W_{lm} \qquad \text{(A19)}$$

$$W_d = min(W_d, W_{dm}) \qquad \text{(A20)}$$

The runoff separation uses the free water capacity curve. The vertical coordinate value ($\beta$)
corresponding to the areal mean free water storage ($S_0$) is:

$$\beta = S_m \times (ex + 1) \times [1 - (1 - S_0/S_m)^{\frac{1}{1+ex}}] \qquad \text{(A21)}$$

Therefore, the surface runoff ($R_s$) is:

$$R_s = \begin{cases} R + \{S_0 - S_m + S_m[1 - \frac{(P_n + \beta)}{S_m \times (ex+1)}]^{ex+1}\}\frac{R}{P_n}, \ P_n + \beta \leq S_m \times (ex + 1) \\ R + (S_0 - S_m)\frac{R}{P_n}, \ P_n + \beta > S_m \times (ex + 1) \end{cases} \qquad \text{(A22)}$$

$$R_s = R_s + R_{imp} \qquad \text{(A23)}$$

The interflow runoff ($R_i$):

$$R_i = K_i \times S_0 \times \frac{R}{P_n} \qquad \text{(A24)}$$

The groundwater runoff ($R_g$):

$$R_g = K_g \times S_0 \times \frac{R}{P_n} \tag{A25}$$


Calculate the areal mean free water storage ($S_0$) at the end of the current period, which will serve as
the initial value for the next period, as:

$$S_0 = S_0 + \left(R - R_s - R_i - R_g\right) \times \frac{P_n}{R} \tag{A26}$$


The flow routing module consists of two submodules: hillslope and channel network routing. The
hillslope routing adopts a linear reservoir approach, while the channel network routing uses the Nash unit
hydrograph. The calculation formula for the linear reservoir is as follows:

$$Q_i = -O_i \times \ln c_i \tag{A27}$$


$$Q_g = -O_g \times \ln c_g \tag{A28}$$


The total inflow to channel network is equal to the sum of the surface runoff ($R_s$), the outflow of
the interflow linear reservoir ($Q_i$), and the outflow of the groundwater linear reservoir ($Q_g$). The
calculation formula is as follows:

$$Q_t = R_s + Q_i + Q_g \tag{A29}$$


The calculation formula for the Nash unit hydrograph reservoir is as follows:

$$Q_1 = F_1/K_f \tag{A30}$$


$$Q_2 = F_2/K_f \tag{A31}$$


$$Q_3 = F_3/K_f \tag{A32}$$


The calculation formula for the outflow discharge of the basin ($Q$) is as follows:

$$Q = Q_3 \times 1000 \times F/\Delta t \tag{A33}$$


where $F$ is the watershed area,km$^2$. $\Delta t$ is the input time step, s.

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
