# Peer review of "An explainable deep learning model based on hydrological principles for flood simulation and"

_EGUsphere, 2025_

## Author Comment (AC2)

**Reply to Reviewers' comments (Reviewer#2)**

This paper integrates the runoff generation and flow routing principles of the Xinanjiang model into a recurrent neural network framework, proposing the XAJRNN layer and constructing an EDL model. This approach enhances the physical interpretability of deep learning-based flood forecasting. Using the Lushui River and Qingjiang River basins as case studies, the EDL model is compared with benchmark models, demonstrating superior performance in flood simulation. The study is well-structured, data-driven, and methodologically rigorous, offering a novel perspective and valuable tool for explainable deep learning in hydrology. However, improvements in clarity, graphical details, and language are needed.

Response: We thank the reviewer for his/her time in reviewing our manuscript and providing comprehensive suggestions for further improvements. Below is our detailed response to the reviewers' comments and suggestions.

(1) Line 128: Please provide additional explanation on why the runoff generation and flow routing principles of the Xinanjiang model were chosen to construct an explainable deep learning model, specifically elaborating on its advantages and applicability.

Response: Thank you for providing this comprehensive review. We chose the runoff generation and flow routing principles of the Xinanjiang (XAJ) model as the foundation for constructing an explainable deep learning model based on the following considerations. First, the XAJ model has demonstrated excellent performance in hydrological simulation and forecasting across various watersheds. Its hydrological principles have been extensively validated over time, ensuring high reliability and maturity. Second, our study area is located in the Yangtze River Basin, which falls within a typical humid and semi-humid climate zone. The XAJ model's saturation excess runoff mechanism effectively captures the nonlinear runoff response under such climatic conditions. This mechanism is particularly suitable for depicting the runoff response of our study area under varying rainfall intensities, thereby providing a solid theoretical foundation for both the interpretability and accuracy of our model.

(2) Line 131: Please further explain the rationale for using LSTM neural network layers to construct the model, highlighting its superiority.

Response: Thank you very much for the notice. We chose LSTM as a component of the explainable deep learning model primarily based on two considerations. First, LSTM's memory units can store hydrological information over long periods, enabling it to effectively model the temporal dependencies in the rainfall-runoff process and enhance flood prediction accuracy. Second, flood evolution involves multiple dynamic processes, including precipitation, evapotranspiration, surface runoff, and groundwater recharge. LSTM can adaptively learn the nonlinear relationships among these variables.

(3) Lines 154-166: To enhance the completeness of the research background, it is recommended that information on the magnitude and frequency of historical floods in the study area be supplemented.

Response: Thank you very much for the notice. We agree with your viewpoint, and we will add information on the magnitude and frequency of historical floods in the background section of the study area. For example, the Qingjiang River Basin experienced major floods in 2016 and 2017, with peak inflow discharge into the Shuibuya Reservoir reaching 13,100 m³/s and 6,710 m³/s, respectively.

**(4) Line 168: Please adjust the scale of the river curves in Figure 1 to improve the aesthetic quality and clarity of the illustration.**

Response: Thank you very much for the notice. We revise the scale of the river curves in Figure 1 to improve the aesthetic quality and clarity. As shown below.

[Figure]

**(5) Line 224: The author mentions "a similar structure"; please specify in which aspects this similarity is reflected to improve clarity.**

Response: Thank you very much for your suggestion. In our manuscript, we mention "a similar structure", which is primarily reflected in the composition of Equations (2) and (3). Both equations consist of two parts: an ordinary differential equation and an output equation, and they share a highly similar structure. Specifically, in the ordinary differential equation part, both equations include the state variable from the previous time step ($h(t-1)$), the state variable at the current time step ($h(t)$), the input ($x$), and the parameters (($\varphi$, $W$, $b$)). In the output equation part, both equations rely on the current state variable ($h(t)$), the output ($y$), and the same set of parameters (($\varphi$, $W$, $b$)).

**(6) Equations (1) and (2) and Figure 3 (b): The parameter symbols in the equations do not match those used in Figure 3 (b). Please carefully verify and ensure consistency.**

Response: Thank you very much for your suggestion. We have reviewed the manuscript and found a minor error. We revise the parameter symbols in Figure 3(b) to ensure consistency with Equations (1) and (2). As shown below.

$$\begin{cases} h(t) = f(h(t-1), x(t); \varphi_f) \\ y(t) = g(h(t), x(t); \varphi_g) \end{cases} \qquad (2)$$

(7) Lines 258-260: The XAJRNN layer outputs four physical variables of interest. Please explain why these four variables were selected as outputs instead of others.

Response: Thank you very much for your suggestion. We chose actual evapotranspiration ($E_t$), areal mean free water storage ($S_0$), areal mean tension water storage ($W$), and outflow discharge ($Q$) as the output variables of the XAJRNN layer, primarily based on their high hydrological relevance to flood forecasting. Actual evapotranspiration ($E_t$) is a key component of the hydrological cycle, directly affecting water availability and being crucial for runoff processes and flood simulation. Areal mean free water storage ($S_0$) and tension water storage ($W$) represent the states of free water and water under tension in the watershed, reflecting the watershed's storage capacity, which in turn influences flood occurrence and intensity. Outflow discharge ($Q$), as the direct output of the basin system, is a core indicator for flood simulating and can directly reflect downstream flood risk. The selection of these variables fully considers their physical significance and practical application value in flood simulation.

(8) Lines 274-280: The paper mentions that a genetic algorithm was used to optimize the parameters of the Xinanjiang model. Please provide the obtained optimal parameter values and include them in the relevant section.

Response: Thank you very much for your suggestion. Below are the calibrated parameter values of the Xinanjiang model.

| Parameter | Value range | Lushui River basin | Qingjiang River basin |
|---|---|---|---|
| $K_c$ | [0.6,1.5] | 0.95 | 0.85 |
| $c$ | [0.01,0.2] | 0.18 | 0.19 |
| $W_{um}$ | [5,30] | 28.75 | 23.15 |
| $W_{lm}$ | [60,90] | 84.36 | 64.47 |
| $W_{dm}$ | [15,60] | 23.19 | 15.60 |
| $A_{imp}$ | [0.01,0.2] | 0.02 | 0.01 |
| $b$ | [0.1,0.4] | 0.40 | 0.35 |
| $S_m$ | [10,50] | 49.97 | 39.86 |
| $ex$ | [1,1.5] | 1.08 | 1.06 |
| $K_i$ | [0.1,0.55] | 0.19 | 0.37 |
| $K_g$ | [0.7-$K_i$] | 0.51 | 0.33 |
| $c_i$ | [0.1,0.9] | 0.87 | 0.89 |
| $c_g$ | [0.9,0.988] | 0.98 | 0.97 |
| $K_f$ | [0.1,10] | 3.99 | 1.58 |

(9) Figure 8 (d): The simulation performance of the EDL and the benchmark models appears to be poor. Please analyze the potential reasons for this issue.

Response: Thank you very much for your suggestion. By analyzing the simulation performance of the EDL model and the benchmark model in Figure 8(d), we identified two major influencing factors: First, the location of the heavy rainfall center has a significant impact on the simulation results. Since the model input uses areal average rainfall, it fails to fully account for the spatial distribution characteristics of rainfall. As shown in Figure 8(d), when the heavy rainfall center is close to the Shuibuya Reservoir, the shortened routing time leads to a significant decline in the model's simulation performance. Second, the impact of upstream reservoir regulation cannot be ignored. During multiple flood events in the Qingjiang River Basin in 2020, the upstream reservoirs of Shuibuya increased their outflow to cope with the severe flood control situation. Combined with the effects of rainfall, this further reduced the model's simulation accuracy.

(10) Language expression: Some parts of the paper contain repetitive phrasing. It is recommended to refine the text to improve fluency and conciseness.

Response: Thank you very much for your suggestion. We will carefully review and refine the language in the manuscript.

(11) Reference formatting: Please carefully check the reference formatting to ensure compliance with the journal's requirements, including the correct spelling of author names, publication year format, DOI, and page ranges.

Response: Thank you very much for your suggestion. We will carefully check the reference format in the manuscript according to the journal's requirements, including names, spelling, and publication years.

---

## Author Response (AR1)

**Cover Letter**

**Dear Editor and Reviewers**

This is the resubmitted version of the revised manuscript entitled: An explainable deep learning model based on hydrological principles for flood simulation and forecasting (Manuscript No. EGUSPHERE-2025-279). The paper has been revised along the lines suggested by the reviewers. All the comments are addressed in the new version of our paper. We have added a table, several references, and more explanation in the revised manuscript. The changed and added parts of the text (except some language correction) are marked in RED color for easy review.

It would be greatly appreciated if the revised version of the paper can be re-evaluated by the same reviewer who spent available time to provide constructive and professional comments and suggestions, which have led to improvement on the presentation and quality of the paper.

We sincerely hope you will find the revised version of the paper is to your satisfaction. We are, of course, more than happy to further improve the paper upon request.

In the following, we provide point to point response to the comments of the reviewers.

Yours sincerely,

May 26, 2025

Corresponding author

Prof. Shenglian Guo

State Key Laboratory of Water Resources Engineering and Management,

Wuhan University, Wuhan 430072, P. R China

**E-mail: slguo@whu.edu.cn**

**Response to Reviewers' comments (Reviewer#1)**

**Legend**

Reviewers' comments

Authors' responses

Direct quotes from the revised manuscript

The paper is well-structured and provides a solid foundation. However, there are a few suggestions for improvement regarding this study:

Response: We thank the reviewer for his/her time in reviewing our manuscript and providing comprehensive suggestions for further improvements. Below is our detailed response to the reviewers' comments and suggestions.

(1) Lines 30–31 mention the limitations of traditional hydrological models. More details on these limitations should be provided.

Response: Thank you for this suggestion. We have revised the introduction section of the manuscript to include a detailed explanation of the limitations of traditional hydrological models:

Lines 30–34: Traditional hydrological models rely on statistical methods and empirical formulas but struggle to accurately simulate complex nonlinear hydrological processes (Roy et al., 2023). Their predefined equations are unable to adapt the climate and environmental changes such as land use and human activities. Additionally, these models often simplify rainfall spatiotemporal distribution and the land surface heterogeneity.

Reference:

Roy, A., Kasiviswanathan, K. S., Patidar, S., Adeloye, A. J., Soundharajan, B. S., and Ojha, C. S. P.: A physics-aware machine learning-based framework for minimizing prediction uncertainty of hydrological models, Water Resour. Res., 59, e2023WR034630, https://doi.org/10.1029/2023WR034630, 2023.

(2) Line 51 contains a typo: "As an RNN subset in DL."

Response: Thank you for pointing this out. We have corrected the typo in Line 51. The revised sentence now reads: "As a subset of RNN in DL, …" to ensure clarity and accuracy. Specifically, as shown below:

Lines 53–54: As a subset of RNN in DL, LSTM has gained prominence for its efficacy in managing sequential data and capturing long-term dependencies.

(3) Figure 1 shows that the rainfall gauge color is too similar to the elevation color. Please use a different color for better distinction. Besides that, the elevations of the two basins can be unified in color scale to make it easier to compare the terrain differences.

Response: Thank you for your valuable suggestions. We have modified the Figure 1 to make the color of the rain gauges more distinguishable. Additionally, we have also unified the color scale for the elevation of the two basins to make it easier to compare the terrain differences. The modified Figure 1 is shown as follows:

[Figure]

Figure 1: Sketch map of river networks and rainfall gauges in the Lushui River and Qingjiang River basins.

(4) In section 2.2, mention that the timestep of data are different in two basins. For example, the rainfall data of Lushui River basin is 3h, but Qingjiang River basin is 6h.

Response: Thank you for your valuable suggestions. We have followed your suggestion and explained the differences in the time steps between the two basins in Section 2.2 as shown below:

Lines 202–203: It should be noted that the time step of these data is 3 h in the Lushui River basin, whereas it is 6 h in the Qingjiang River basin.

(5) The Xinanjiang (XAJ) hydrological model should be explicitly mentioned in Line 182.

Response: Thank you for your suggestion. We have provided more detailed explanation of the Xinanjiang model in the revised version:

Lines 212–221: Zhao (1992, 1993) firstly proposed the XAJ model for rainfall-runoff simulation and flood forecasting in the 1970s, and it is a classic conceptual hydrological model and has been widely used in China. The core concept of the XAJ model is the runoff formation on repletion of storage: runoff is not produced until the soil moisture content of the aeration zone reaches the field capacity. Once this threshold is exceeded, all additional rainfall is converted directly into runoff without further loss. Runoff production at a specific point occurs only after the tension water storage at that point is fully saturated. To represent the spatial heterogeneity of tension water capacity within a basin, the model introduces a tension water capacity curve. In terms of runoff separation, the model, based on empirical observations and theoretical studies, adds an additional component: interflow, on top of the original division into surface runoff and groundwater runoff (Yao et al., 2009, 2014).

Reference:

Zhao, R.: The Xinanjiang model applied in China, J. Hydrol., 135, 371–381, https://doi.org/10.1016/0022-1694(92)90096-E, 1992.

Zhao, R.: A non-linear system model for basin concentration, J. Hydrol., 142, 477–482, https://doi.org/10.1016/0022-1694(93)90024-4, 1993.

Yao, C., Li, Z., Bao, H., and Yu, Z.: Application of a developed grid-Xinanjiang Model to Chinese watersheds for flood forecasting purpose, J. Hydrol. Eng., 14, 923–934, https://doi.org/10.1061/(ASCE)HE.1943-5584.0000067, 2009.

Yao, C., Zhang, K., Yu, Z., Li, Z., and Li, Q.: Improving the flood prediction capability of the Xinanjiang model in ungauged nested catchments by coupling it with the geomorphologic instantaneous unit hydrograph, J. Hydrol., 517, 1035–1048, https://doi.org/10.1016/j.jhydrol.2014.06.037, 2014.

(6) Equation 2 and Figure 3 present the proposed XAJRANN layer. How are its parameters optimized?

Response: Thank you for your question. Following the general optimization methods of deep learning models, the parameters of the XAJRANN layer are optimized using gradient descent, specifically with the Adam optimizer to minimize the loss function. We have added the corresponding explanation in the revised manuscript:

Lines 352–355: Following the general optimization methods of DL models, the parameters of the XAJRANN and LSTM layer in the EDL model are optimized using gradient descent, specifically with the *Adam* optimizer, to minimize the loss function. The model is trained with NSE as the loss function and a learning rate of 0.001.

(7) Lines 247-248 mention that "The choice of LSTM is based on the numerous studies demonstrating its ability to improve the performance of hydrological model simulations.", the author should add some references for these statements.

Response: Thank you for the reviewer's suggestion. To support the statement regarding the choice of LSTM in lines 247-248, we have added some relevant references. These studies demonstrated that LSTM performs well and can effectively improve simulation accuracy:

Lines 320–329: The choice of LSTM layers is based on two primary considerations: first, its memory cells can retain hydrological information over extended periods, effectively capturing the temporal dependencies of the rainfall–runoff process to enhance flood simulating accuracy; and second, many studies have demonstrated that LSTM consistently improves hydrological model simulation performance. For example, Alizadeh et al. (2021) demonstrated the SAINA-LSTM model outperforms the EnsPost and MS-EnsPost in low, medium, and high flow ranges, as well as in 1 to 7 day forecast horizons, and significantly reduces the root mean square error of flow predictions. Additionally, Xu et al. (2022) combined the particle swarm optimization (PSO) algorithm with the LSTM model to obtain the PSO-LSTM model. The research results show that the PSO-LSTM model outperforms the artificial neural network (ANN) and PSO-ANN at all stations in the basin.

Reference:

Alizadeh, B., Ghaderi Bafti, A., Kamangir, H., Zhang, Y., Wright, D. B., and Franz, K. J.: A novel attention-based LSTM cell post-processor coupled with Bayesian optimization for streamflow prediction, J. Hydrol., 601, 126526,

https://doi.org/10.1016/j.jhydrol.2021.126526, 2021.

Xu, Y., Hu, C., Wu, Q., Jian, S., Li, Z., Chen, Y., Zhang, G., Zhang, Z., and Wang, S.: Research on particle swarm optimization in LSTM neural networks for rainfall-runoff simulation, J. Hydrol., 608, 127553, https://doi.org/10.1016/j.jhydrol.2022.127553, 2022.

(8) Figure 4 illustrates the structure of the EDL model. In the XAJRNN layer, the model outputs actual evapotranspiration (Et), areal mean free water storage (S0), areal mean tension water storage (W), and basin outflow discharge (Q). Is this output generated through supervised learning? Subsequently, in the LSTM layer, the model produces runoff, which may also be trained using supervised learning. Does it make sense to train the model twice?

Response: Thank you for your question. In our model, the training of the XAJRNN layer and the LSTM layer is complementary and not an independent "two-step training" process. The XAJRNN layer simulates actual evapotranspiration ($E_t$), water storage ($S_0$, W), and basin outflow (Q) based on input data through supervised learning. These intermediate outputs then serve as the input for the LSTM layer, which further processes them to predict runoff. The training process of the entire model is integrated, and the parameters of both the XAJRNN and LSTM layers are jointly optimized to improve prediction accuracy. Therefore, even though there are multiple layers of output, their training is interconnected, and there is no issue of redundant training. We have further clarified the training process of the model, and explained how the XAJRNN and LSTM layers work together to optimize overall performance in the revised manuscript.

(9) Is there a typo in Lines 296 "and vis vasa."? (vice versa?)

Response: Thank you for your careful observation. Indeed, "and vis vasa" in line 296 is a typo, and it should be "vice versa." We have corrected this error in the revised manuscript.

Lines 458–459: If $\Delta T$ is positive, the simulated peak discharge occurs early than the observed peak discharge; and vice versa.

(10) The RMSE values of the two basins in Table 1 are quite different. Please explain the results based on data statistics.

Response: Thank you for your question. The significant difference in RMSE between the two basins is primarily due to the disparity in their annual average flows. Based on statistical calculations, the annual average flow of the Qingjiang Basin is 290 m³/s, while that of the Lushui Basin is only 96 m³/s. Additionally, the overall simulation performance in the Lushui Basin is better than in the Qingjiang Basin, which contributes to the higher RMSE observed in the Qingjiang Basin. We have included the corresponding analysis in the revised manuscript:

Lines 404–409: It is important to note that the *RMSE* values of the two basins in Table 1 differ significantly. Specifically, the *RMSE* in the Lushui River basin is noticeably lower than that in the Qingjiang River basin. A possible reason for this difference is that, based on statistical calculations, the annual average flow of the Qingjiang River basin is 290 m³/s, whereas that of the Lushui River basin is only 96 m³/s. Additionally, the overall simulation performance in the Lushui River basin is better than in the Qingjiang River basin.

(11) Table 1 presents the model performance. During the testing phase, the LSTM model achieved better NSE, RE, and RMSE for the Qingjiang River. What could be the reason for this?

Response: Thank you for your questions. During the testing phase, the LSTM model demonstrated better simulation performance in the Qingjiang Basin, primarily due to the close integration of our explainable deep learning (EDL) model with the Xinanjiang model. Specifically, the XAJRNN layer in the EDL model adopts the runoff generation and routing principles of the XAJ model, making the model performance closely related to the simulation accuracy of the XAJ model. When the XAJ model performs well, the EDL model also improves accordingly, vice versa. We have included the corresponding analysis in the revised manuscript:

Lines 409–414: During the test phase, the LSTM model demonstrated better simulation performance in the Qingjiang River basin, primarily due to the close integration of our EDL model with the XAJ model. Specifically, the XAJRNN layer in the EDL model adopts the runoff generation and routing principles of the XAJ model, and the model performance is closely related to the simulation accuracy of the XAJ model. When the XAJ model performs well, the EDL model also achieves better *NSE*, *RE*, and *RMSE* for the Qingjiang River.

(12) What do the colors in the scatter plots of Figure 5 and Figure 6 represent? Please add a legend.

Response: Thank you for your suggestion. The colors in Figures 5 and 6 represent the density of scatter point distribution, where higher density corresponds to higher color intensity. To clarify this, we have added a legend to explicitly indicate the meaning of the colors in the revised manuscript.

Lines 446–449:

[Figure]

(a) EDL model

(b) XAJ model

(c) LSTM model

Figure 5: Scatter plots of observed (Qo) and simulated (Qp) flow discharges by three models in the Lushui River basin. The color bar represents the density of the scatter distribution. The denser the scatter distribution, the higher the corresponding density value in color.

(13) Figure 5 shows that there are flood events exceeding 3000 m³/s during the test period, while there are fewer flood events exceeding 3000 m³/s during the training period. Please explain whether this is the reason why the scatter points are below the 1:1 ideal line in the high flow range.

Response: Thank you for your suggestion. We agree with your point that multiple flood events with flow rates exceeding 3,000 m³/s occurred during the testing period, while such events were relatively rare during the training period. This indicates that the model had limited training in the high-flow range, leading to scatter points in this range being positioned below the 1:1 ideal line. We have added the corresponding analysis in the conclusion section:

Lines 428–432: The reason that the scatter points fall below the 1:1 ideal line in the high flow range may be due to the fact that during the training period, there were few flow values exceeding 3000 m³/s, while in the test period, there were relatively more high flows exceeding 3000 m³/s.

(14) Table 2 indicates that the $\Delta T$ of XAJ and LSTM model are more than one day in 20170702 event. Is there a calculation error since the NSE is 0.93 for LSTM?

Response: Thank you for your question. In the 20170702 flood event, the $\Delta T$ of both the XAJ model and the LSTM model exceeded one day, yet the NSE of the LSTM model still reached 0.93. This is not a calculation error, and the reasons are as follows: As shown in Figure 7(b), the 20170702 flood event was a typical double-peak flood. The LSTM model overestimated the first peak, resulting in a $\Delta T$ of more than one day, a phenomenon also observed in the XAJ model. However, the overall flood hydrograph simulated by the LSTM model closely matches the observed hydrograph, leading to a high NSE of 0.93. In contrast, the XAJ model's simulated hydrograph deviated more from the observed hydrograph, resulting in a lower NSE of only 0.65.

(15) There is a problem with the statements for Lines 394-396. The discrepancies in the rising speed during the flood rising phase compared to the observations may be due to the slow response

of the model to rainfall rather than to the models' insufficient ability to simulate low flow conditions.

Response: Thank you for your valuable comments. We agree with your viewpoint that the discrepancy in the rising speed during the flood rising phase may be related to the model's response speed to rainfall, rather than solely due to its ability to simulate low flow conditions. We have revised this section in the manuscript.

Lines 506–507: This may be due to the slow response of the model to rainfall.

(16) Lines 401-403 mention that all three models underestimated the peak flow, and the simulated peak was significantly delayed compared to the observed peak, especially under complex terrain conditions. Please select stations with complex terrain and simple terrain for result comparisons to illustrate the impact of terrain on model simulations.

Response: Thank you for your suggestion. In the manuscript, we mentioned that all three models underestimated the peak flow and there was a delay in the simulated peak. This is because our study involved two basins, the Lushui and Qingjiang basins. As shown in Figure 1, the Qingjiang Basin has a more complex terrain and a more meandering river network. Based on the flood simulation results in Figures 7 and 8, the simulation accuracy in the Lushui Basin is better than that in the Qingjiang Basin. This led us to conclude that under complex terrain conditions, the model might not perform as well as under simple terrain conditions. We have added the above analysis in the relevant sections.

Lines 520–525: Our study focused on two basins: the Lushui and Qingjiang basins. As illustrated in Figure 1, the Qingjiang Basin features a more complex terrain and a more meandering river network compared to the Lushui Basin. Based on the flow simulation results shown in Figures 7 and 8, the model performance in the Lushui Basin is better than that in the Qingjiang Basin. These findings suggest that the model simulation accuracy in simple terrain basin is higher than that in the complex terrain conditions.

(17) The author should add some statements about the simulated time horizon (e.g. T+1, T+2, …).

Response: Thank you for your suggestion. We acknowledge the concept of "simulated time horizon" as mentioned. However, our study pertains to flood simulation rather than forecasting, and thus does not involve forecast horizons. Consequently, expressions like T+1, T+2, etc., are not used in the manuscript.

**Reply to Reviewers' comments (Reviewer#2)**

This paper integrates the runoff generation and flow routing principles of the Xinanjiang model into a recurrent neural network framework, proposing the XAJRNN layer and constructing an EDL model. This approach enhances the physical interpretability of deep learning-based flood forecasting. Using the Lushui River and Qingjiang River basins as case studies, the EDL model is compared with benchmark models, demonstrating superior performance in flood simulation. The study is well-structured, data-driven, and methodologically rigorous, offering a novel perspective and valuable tool for explainable deep learning in hydrology. However, improvements in clarity, graphical details, and language are needed.

Response: We thank the reviewer for his/her time in reviewing our manuscript and providing comprehensive suggestions for further improvements. Below is our detailed response to the reviewers' comments and suggestions.

(1) Line 128: Please provide additional explanation on why the runoff generation and flow routing principles of the Xinanjiang model were chosen to construct an explainable deep learning model, specifically elaborating on its advantages and applicability.

Response: Thank you for providing this comprehensive review. We chose the runoff generation and flow routing principles of the Xinanjiang (XAJ) model as the foundation for constructing an explainable deep learning model based on the following considerations. First, the main feature of the XAJ model is runoff formation on repletion of storage. Second, the flow routing of the XAJ model includes hillslope routing and channel network routing. We have added a detailed explanation of this issue in Section 1, **Introduction**:

Lines 150–155: The XAJ model proposed by Zhao (1992, 1993) has been widely used for hydrological simulation and flood forecasting in China. The flow routing of the XAJ model includes hillslope routing and channel network routing (Yao et al., 2009, 2014), which are represented by linear reservoirs and Nash unit hydrographs (Singh, 1977), respectively. Compared with other lumped hydrological models, the XAJ model performs very well in the humid and semi-humid regions, which helps to better highlight the distinctive aspects of this study.

Reference:

Singh, V. P.: Estimation of parameters of a uniformly nonlinear surface runoff model, Hydrol. Res., 8, 33–46, https://doi.org/10.2166/nh.1977.0003, 1977.

(2) Line 131: Please further explain the rationale for using LSTM neural network layers to construct the model, highlighting its superiority.

Response: Thank you very much for the notice. We selected the LSTM layer for our EDL model based on two primary considerations: first, its memory cells can retain hydrological information over extended periods, effectively capturing the temporal dependencies of the rainfall–runoff process to enhance flood forecasting accuracy; and second, numerous studies have demonstrated that LSTM consistently improves hydrological model simulation performance. We have added the corresponding content in Section 3.2.1 **EDL model**:

Lines 320–324: The choice of LSTM layers is based on two primary considerations: first, its memory cells can retain hydrological information over extended periods, effectively capturing the temporal dependencies of the rainfall–runoff process to enhance flood simulating accuracy; and

second, many studies have demonstrated that LSTM consistently improves hydrological model simulation performance.

(3) Lines 154-166: To enhance the completeness of the research background, it is recommended that information on the magnitude and frequency of historical floods in the study area be supplemented.

Response: Thank you very much for the notice. We agree with your viewpoint, and we have added information on the magnitude and frequency of historical floods in the background section of the study area:

Lines 192–194: The Qingjiang River basin experienced major floods in 2016 and 2017, with peak inflow discharge of the Shuibuya Reservoir reaching 13,100 m³/s and 6,710 m³/s, respectively.

(4) Line 168: Please adjust the scale of the river curves in Figure 1 to improve the aesthetic quality and clarity of the illustration.

Response: Thank you very much for the notice. We have revised the scale of the river curves in Figure 1 to improve the aesthetic quality and clarity.

(5) Line 224: The author mentions "a similar structure"; please specify in which aspects this similarity is reflected to improve clarity.

Response: Thank you very much for your suggestion. In our manuscript, we mention "a similar structure", which is primarily reflected in the composition of Equations (2) and (3). Both equations consist of two parts: an ordinary differential equation and an output equation, and they share a highly similar structure. Specifically, in the ordinary differential equation part, both equations include the state variable from the previous time step (h(t-1)), the state variable at the current time step (h(t)), the input (x), and the parameters ((φ, W, b)). In the output equation part, both equations rely on the current state variable (h(t)), the output (y), and the same set of parameters ((φ, W, b)). We have added the corresponding content of the manuscript.

Lines 263–268: Both equations consist of two parts: an ordinary differential equation and an output equation, and they share a highly similar structure. Specifically, in the ordinary differential equation part, both equations include the state variable from the previous time step (h(t-1)), the state variable at the current time step (h(t)), the input (x), and the parameters ((φ, W, b)). In the output equation part, both equations rely on the current state variable (h(t)), the output (y), and the same set of parameters ((φ, W, b)).

(6) Equations (1) and (2) and Figure 3 (b): The parameter symbols in the equations do not match those used in Figure 3 (b). Please carefully verify and ensure consistency.

Response: Thank you very much for your suggestion. We have reviewed the manuscript and found a minor error. We have revised the parameter symbols in Figure 3(b) to ensure consistency with Equations (1) and (2).

Lines 308–309:

[Figure]

(a) Ordinary RNN unit           (b) Proposed XAJRNN unit

Figure 3: The structures of ordinary RNN unit and proposed XAJRNN unit.

(7) Lines 258-260: The XAJRNN layer outputs four physical variables of interest. Please explain why these four variables were selected as outputs instead of others.

Response: Thank you very much for your suggestion. We chose actual evapotranspiration ($E_t$), areal mean free water storage ($S_0$), areal mean tension water storage (W), and outflow discharge (Q) as the output variables of the XAJRNN layer, primarily based on their high hydrological relevance to flood forecasting. Actual evapotranspiration ($E_t$) is a key component of the hydrological cycle, directly affecting water availability and being crucial for runoff processes and flood prediction. Areal mean free water storage ($S_0$) and tension water storage (W) represent the states of free water and water under tension in the watershed, reflecting the watershed's storage capacity, which in turn influences flood occurrence and intensity. Outflow discharge (Q), as the direct output of the watershed system, is a core indicator for flood forecasting and can directly reflect downstream flood risk. The selection of these variables fully considers their physical significance and practical application value in flood forecasting. We have added the relevant explanation in Section 3.2.1, **EDL model**, of the manuscript:

Lines 341–348: The selection of these four variables as the output of the XAJRNN layer is primarily based on their high hydrological relevance to flood forecasting. Actual evapotranspiration ($E_t$) is a key component of the hydrological cycle, directly affecting water availability and playing a crucial role in runoff processes and flood simulation. Areal mean free water storage ($S_0$) and tension water storage ($W$) represent the states of free water and water under tension in the watershed, reflecting the basin storage capacity, which in turn influences flood occurrence and intensity. Outflow discharge ($Q$), as the direct output of the basin system, is a core indicator for flood simulation. The selection of these variables fully considers their physical significance and practical application value in flood simulation.

(8) Lines 274-280: The paper mentions that a genetic algorithm was used to optimize the parameters of the Xinanjiang model. Please provide the obtained optimal parameter values and include them in the relevant section.

Response: Thank you very much for your suggestion. We have added the calibrated parameter values of the Xinanjiang model to the revised manuscript.

Lines 375-376:

**Table 1 Optimal parameters of the ordinary XAJ model calibrated using the GA algorithm.**

| Parameter | Value range | Lushui River basin | Qingjiang River basin |
|:---:|:---:|:---:|:---:|
| $K_c$ | [0.6,1.5] | 0.95 | 0.85 |
| $c$ | [0.01,0.2] | 0.18 | 0.19 |
| $W_{um}$ | [5,30] | 28.75 | 23.15 |
| $W_{lm}$ | [60,90] | 84.36 | 64.47 |
| $W_{dm}$ | [15,60] | 23.19 | 15.60 |
| $A_{imp}$ | [0.01,0.2] | 0.02 | 0.01 |
| $b$ | [0.1,0.4] | 0.40 | 0.35 |
| $S_m$ | [10,50] | 49.97 | 39.86 |
| $ex$ | [1,1.5] | 1.08 | 1.06 |
| $K_i$ | [0.1,0.55] | 0.19 | 0.37 |
| $K_g$ | [0.7-$K_i$] | 0.51 | 0.33 |
| $c_i$ | [0.1,0.9] | 0.87 | 0.89 |
| $c_g$ | [0.9,0.988] | 0.98 | 0.97 |
| $K_f$ | [0.1,10] | 3.99 | 1.58 |

(9) Figure 8 (d): The simulation performance of the EDL and the benchmark models appears to be poor. Please analyze the potential reasons for this issue.

Response: Thank you very much for your suggestion. By analyzing the simulation performance of the EDL model and the benchmark model in Figure 8(d), we identified two major influencing factors: First, the location of the heavy rainfall center has a significant impact on the simulation results. Second, the impact of upstream reservoir regulation cannot be ignored. The detailed explanation have been incorporated into the revised manuscript:

Lines 511-518: The poor simulation performance may be attributed to two major influencing factors. First, the location of the heavy rainfall center has a significant impact on the simulation results. Since the model input uses areal average rainfall, it fails to fully account for the spatial distribution characteristics of rainfall. As shown in Figure 8(d), when the heavy rainfall center is close to the Shuibuya Reservoir, the short flow routing time leads to a significant decline in the model simulation performance. Second, the impact of upstream reservoir regulation cannot be ignored. During multiple flood events in the Qingjiang River basin in 2020, the Shuibuya Reservoir increased outflow discharge to cope with severe flood control conditions.

(10) Language expression: Some parts of the paper contain repetitive phrasing. It is recommended to refine the text to improve fluency and conciseness.

Response: Thank you very much for your suggestion. We have carefully reviewed and refined the language in the manuscript.

(11) Reference formatting: Please carefully check the reference formatting to ensure compliance with the journal's requirements, including the correct spelling of author names, publication year format, DOI, and page ranges.

Response: Thank you very much for your suggestion. We have carefully checked the reference format in the manuscript according to the journal's requirements, including names, spelling, and publication years.

**Response to Reviewers' comments (Reviewer#3)**

**Legend**

Reviewers' comments

Authors' responses

Direct quotes from the revised manuscript

This manuscript introduces an explainable deep learning (EDL) model that merges the Xin'anjiang (XAJ) hydrological model's principles with recurrent neural network (RNN) units for flood simulation. The EDL model, tested in two Chinese river basins, outperforms benchmark models with high accuracy, demonstrating how incorporating physical constraints into deep learning architectures can enhance both simulation accuracy and model interpretability for hydrological modeling. These findings contribute to improved flood forecasting capabilities, offering a promising approach for combining traditional hydrological knowledge with advanced machine learning techniques. But several critical issues require attention.

Response: We thank the reviewer for his/her time in reviewing our manuscript and providing comprehensive suggestions for further improvements. Below is our detailed response to the reviewers' comments and suggestions.

Points for the Authors to Consider

1. Novelty and Positioning within Existing Literature

The integration of physical models with deep learning has evolved rapidly, with existing approaches broadly categorized into loose coupling (e.g., using physical constraints in loss functions, post-processing physical model outputs with DL) and tight coupling (e.g., embedding physical equations into neural network architectures) mentioned in this manuscript. While the manuscript briefly mentions some loose coupling methods, it overlooks key advancements in both paradigms. For instance, prior studies have initialized DL model weights using physical simulations (Read et al., 2019; Yang et al., 2020), combined offline-trained DL modules with physics-based models (Li et al., 2014). For tight coupling frameworks, differentiable models (DM) (Shen et al., 2023) demonstrate how end-to-end training enables deep integration of physical and neural components, going beyond parameter learning to include module replacement and hybrid architectures. This spectrum of tight coupling approaches is further illustrated by physics-embedded models like the Mass-Conserving LSTM (Hoedt et al., 2021) and RNNs with Mass-Conserving-Perceptron (MCP) (Wang and Gupta, 2024), which directly incorporate conservation laws into network architectures.

Another innovative approach involves using physics-based equations as the fundamental time units within RNN structures (He et al., 2024). Notably, many contemporary studies employ hybrid coupling strategies that combine both loose and tight coupling elements. For instance, DM frameworks frequently utilize parameter learning while simultaneously replacing specific physics-based modules with neural network subcomponents (Feng et al., 2022). For this manuscript, the proposed EDL model appears functionally similar to existing hybrid approaches, particularly the physics-embedded models with some neural network modules to postprocess the results (Jiang et al., 2020). Without a clearer differentiation from these methods—such as

explicating how the XAJRNN layer uniquely preserves hydrological parameters or interacts with LSTM modules—the

claimed novelty remains ambiguous. A more thorough literature review and direct comparisons with recent physics-guided DL frameworks would better contextualize the work's contributions.

Response: Thank you for your valuable comments on this study. We greatly appreciate your suggestions regarding the positioning of our research within the existing literature and its originality. In response to your advice, we have revised the manuscript accordingly and added relevant references to provide a more comprehensive review of recent advances in the coupling of physical principles with deep learning. We have also further clarified the uniqueness of our proposed approach, as detailed below:

Lines 96-104: Yang et al. (2020) proposed a hybrid modeling framework that integrates a physically distributed hydrological model (GBHM), artificial neural networks (ANN), a categorization approach (CA), and computer vision (CV) to enhance hydrological simulations in data-scarce watersheds. They show that these models can significantly improve the ability to capture spatial variability and simulate extreme flow events. Li et al. (2014) proposed a black-box model which combines the back-propagation neural network (BPNN) with the K-nearest neighbor (KNN) algorithm, and developed two hybrid models (XBK and XSBK) by coupling the black-box routing module with the runoff generation and separation modules of the XAJ model. Applications in multiple watersheds demonstrate that these hybrid models outperform both the traditional BPNN and the XAJ model.

Lines 124-128: Wang and Gupta (2024) explored the use of mass-conserving perceptron (MCP)-based directed graph architectures to develop minimal and interpretable hydrological model structures within a single catchment. They found that this framework significantly enhances flow simulation performance, particularly when augmented with input bypass and bi-directional groundwater exchange mechanisms.

Lines 136-139: Similarly, He et al. (2024) proposed a deep process learning (DPL) approach, which allows neural networks to infer underlying process mechanisms from observational data by embedding intuitive physical laws of geosystems directly into the DL architecture as structural priors.

Lines 144-149: Even the more advanced methods which embed physical mechanisms directly into neural network layers, they are relied on relatively simple or empirical physical models. To achieve real breakthroughs in hydrological forecasting, it is still necessary to systematically integrate more complex hydrological physical processes into neural network architectures, thereby ensuring both rigorous physical interpretability and superior forecasting performance under future scenarios.

Lines 155-158: The core innovation of this study is the development of a novel XAJRNN layer that converts the XAJ model's sophisticated runoff generation and flow routing mechanisms into differential equation form and embeds them within a conventional RNN unit framework by explicitly defining its state variables and fluxes.

Reference:

Yang, S., Yang, D., Chen, J., Santisirisomboon, J., Lu, W., and Zhao, B.: A physical process and machine learning combined hydrological model for daily streamflow simulations of large watersheds with limited observation data, J. Hydrol., 590, 125206, https://doi.org/10.1016/j.jhydrol.2020.125206, 2020.

Li, Z., Kan, G., Yao, C., Liu, Z., Li, Q., and Yu, S.: Improved neural network model and its

application in hydrological simulation, J. Hydrol. Eng., 19, 04014019, https://doi.org/10.1061/(ASCE)HE.1943-5584.0000958, 2014.

Wang, Y.-H. and Gupta, H. V.: Towards interpretable physical-conceptual catchment-scale hydrological modeling using the mass-conserving-perceptron, Water Resour. Res., 60, e2024WR037224, https://doi.org/10.1029/2024WR037224, 2024.

He, L., Shi, L., Song, W., Shen, J., Wang, L., Hu, X., and Zha, Y.: Synergizing intuitive physics and big data in deep learning: Can we obtain process insights while maintaining state-ff-the-art hydrological prediction capability?, Water Resour. Res., 60, e2024WR037582, https://doi.org/10.1029/2024WR037582, 2024.

2. Methodological Transparency and Reproducibility

The manuscript lacks critical details about the XAJRNN implementation. For example, the mathematical formulation linking Nash unit hydrographs to neural network operations (Appendix A, Eqs. A30–A32) is insufficient to reconstruct the model. Key questions remain unresolved: How are XAJ parameters (e.g., tension water capacity) represented in the hybrid architecture? What specific components of the XAJ model are preserved versus replaced by DL modules? Without open-source code or a complete algorithmic description, the reproducibility and reliability of the results are compromised.

Response: Thank you very much for your valuable comments. We fully agree with your point regarding the lack of critical implementation details of the XAJRNN in the manuscript. Therefore, we have supplemented several sections using the equations in the appendix and pseudocode to illustrate the implementation details of XAJRNN. In these additions, we focus on how the specific components and equations of the XAJ model are applied within the XAJRNN unit, detailing the intermediate processes, variables, and corresponding parameters. The details are as follows:

Lines 278-307: The internal computation process of the XAJRNN unit is explained below using pseudocode and equations provided in the appendix. For each XAJRNN unit, it is first necessary to initialize the watershed state, including the areal mean tension water storage of the upper ($W_u$), lower ($W_l$), and deep ($W_d$) soil layers of the watershed, the areal mean free water storage ($S_0$), the storage of the interflow linear reservoir ($O_i$), the storage of the groundwater linear reservoir ($O_g$), and the storage of the three reservoirs in the Nash unit hydrograph ($F_1$, $F_2$ and $F_3$). Then, the hydrological response of the XAJRNN unit at each time step is carried out through a "step_function", in which all computations are encapsulated. The network calls the "step_function" in a sequential manner. The function mainly includes the following four sub-functions:

(1)Based on the three-layer soil moisture model and the tension water capacity curve, the runoff generation from the permeable portion of the watershed is calculated. The corresponding equations are A1 to A12, and the main parameters involved include $A_{imp}$,$K_c$,$c$,$b$,$W_{um}$,$W_{lm}$, and $W_{dm}$. Then, the areal mean tension water storage of the upper ($W_u$), lower ($W_l$), and deep ($W_d$) soil layers of the watershed is updated, which will serve as the initial values for the next period. This corresponds to equations A13 to A20. The main outputs obtained in this sub-function are the runoff ($R$), actual evapotranspiration ($E_t$), and net rainfall ($P_n$).

(2) The runoff ($R$) is divided into different components using the free water capacity curve. The corresponding equations are A21 to A25, and the parameters involved include $S_m$,$ex$,$K_i$, and $K_g$. Then, the areal mean free water storage ($S_0$) is updated according to equation A26. The outputs obtained in this sub-function are surface runoff ($R_s$), interflow runoff ($R_i$), and groundwater runoff

$(R_g)$.

(3) The interflow runoff $(R_i)$ and groundwater runoff $(R_g)$ obtained earlier are routed over the hillslope using the linear reservoir method. The corresponding equations are A27 and A28, with parameters $c_i$ and $c_g$. Then, combined with the surface runoff $(R_s)$, the total inflow $(Q_t)$ into the channel network is calculated, corresponding to equation A29. The outputs obtained are the outflow of the interflow linear reservoir $(Q_i)$, the outflow of the groundwater linear reservoir $(Q_g)$, and total inflow $(Q_t)$.

(4) The Nash unit hydrograph method is used to perform channel network routing, resulting in the final outflow $(Q)$. The corresponding equations are A30 to A33, and the parameter involved is $K_f$. All of the above physical parameters are automatically adjusted during the model training process through gradient descent and backpropagation.

The post-processing modules (normalization and LSTM layers in Figure 4) also warrant deeper analysis. Their impact on model performance—particularly in extreme events where the EDL model shows better performance (Tables 1–2)—is unclear. Ablation studies comparing the full EDL model to standalone XAJRNN and postprocessing components would clarify their relative contributions.

Response: Thank you very much for your suggestions regarding the post-processing components. Our approach is as follows. In our study, we set up two benchmark models: an ordinary XAJ model and an LSTM model. The LSTM model's architecture can be used to compare the relative contribution of the post-processing components in the EDL model because the LSTM model differs from the EDL model only by the absence of the XAJRNN layer, while the rest remains the same. The ordinary XAJ model serves as the standalone XAJRNN layer to illustrate its contribution since, as mentioned earlier, the main components of the XAJRNN layer are the same as those of the ordinary XAJ model. In the revised manuscript, we have added corresponding explanations to clarify this:

Lines 359-363: The first benchmark model is the LSTM model, which differs from the EDL model only in the absence of the XAJRNN layer; the rest of the architecture, including the normalization layer and LSTM layer, remains the same. The purpose of this model is to compare the contribution of the post-processing components to the simulation performance in the EDL model.

Lines 367-369: The second benchmark model is the ordinary XAJ model, which also takes rainfall and evaporation as input to simulate outflow discharge. The purpose of this model is to demonstrate the contribution of the XAJRNN layer in the EDL model.

3. Validation and Generalizability

The evaluation limited to two basins raises concerns about generalizability. Deep learning models typically require diverse datasets to demonstrate robustness, yet the study does not test the EDL model in data-scarce regions or basins with contrasting hydrological regimes. Expanding the validation to additional basins and explicitly analyzing performance across flood magnitudes (e.g., low, medium, and extreme flows) would strengthen the claims.

Response: Thank you for your suggestions regarding validation and generalizability. We fully agree with your point that deep learning models typically require diverse datasets to demonstrate their robustness. Evaluating the EDL model in only two basins may indeed raise concerns about its generalizability. However, considering the difficulty in obtaining data and the

representativeness of the two selected basins within the Yangtze River Basin in China. In our future work, we will follow your recommendation to evaluate the performance of the EDL model under varying watershed conditions and flood magnitudes.

Specific Comments

1. Line 43 contains a typographical error where "deep." appears. This should be corrected to "deep".

Response: Thank you very much for your suggestion. The corrections have already been made in our revised manuscript.

Lines 44-46: The term "deep" in DL signifies network structures with multiple layers and neurons, although there is no precise definition of "deep".

2. In lines 78-81, the final example in this paragraph would benefit from improved readability. Adding a transitional conjunction like "but" before the final point would enhance the flow of the argument.

Response: Thank you very much for your suggestion. Following your suggestion, we have revised the manuscript.

Lines 80-84: Frame et al. (2023) concluded that strictly adhering to the principle of water balance may reduce forecasting performance due to data errors. However, DL models do not require the enforcement of the water balance principle and can adapt to data biases, outperforming traditional hydrological models in runoff forecasting.

3. At line 90, where the abbreviation "EDL" first appears in the manuscript, it should be properly introduced by providing its full name, "explainable deep learning", before using the abbreviated form.

Response: Thank you very much for your suggestion. This was our mistake — it should be a hybrid model rather than an EDL model. We have corrected it in the revised manuscript.

Lines 93-95: Cui et al. (2021) proposed a novel hybrid model combining the Xinanjiang (XAJ) hydrological model with LSTM for multi-step flood forecasting.

4. Line 301 contains a subject-verb agreement error in the phrase "The evaluation metrics includes". The verb should be corrected to "include".

Response: Thank you very much for your suggestion. Following your suggestion, we have revised the manuscript.

Lines 397-398: The evaluation metrics include NSE, RE, and RMSE values for both training and test phases.

5. The reference to "high flow range" at line 322 lacks quantitative specificity. This should be made more precise by providing a numerical threshold, such as "flows larger than [specific value] $m^3/s$".

Response: Thank you very much for your suggestion. Following your suggestion, we have specified the exact range of high flow values in the revised manuscript:

Lines 421-422: However, during the test period, in the range where observed flow exceeds 3000 $m^3/s$, most scatter points are located below the 1:1 ideal line.

6. Several instances throughout the text (lines 335, 339, 340, 342, 390, 391, 393, 404, 407, 409) incorrectly use "DEL" when referring to the model. These should all be corrected to "EDL" for consistency. A thorough check of the entire manuscript for this error is recommended.
Response: Thank you very much for your suggestion. This was a writing error on our part. We have reviewed the entire manuscript and made the necessary corrections.

7. Tables 2 and 3 present negative values for the $\Delta TT$ (timing error) metric without sufficient explanation. The authors should clarify the hydrological significance of these negative values in the context of flood peak timing predictions.
Response: Thank you very much for your suggestion. Following your suggestion, We have included the explanation in the revised manuscript:

Lines 458-459: It should be specifically noted that if $\Delta T$ is positive, the simulated peak discharge occurs earlier than the observed peak discharge; and vice versa.

8. The manuscript currently lacks proper reference to Tables 2 and 3 in the text. When first discussing results from these tables, explicit references such as "as shown in Table 2" or "see Table 3" should be included to guide readers.
Response: Thank you very much for your suggestion. Following your suggestion, We have revised the content in the manuscript:

Line 477: As shown in Table 2, the EDL model performed exceptionally well with high simulation accuracy,

Line 485: As shown in Table 3, the EDL model continued to demonstrate superior performance,

9. At line 388, there appears to be confusion between "XAJRNN" and "XAJ". These are distinct entities and should not be used interchangeably. The text should clarify their relationship and differences.
Response: Thank you for your suggestion. This was a writing error on our part, and it has been corrected in the revised manuscript:

Lines 497-498: comparing the simulation results of the EDL model, XAJ model, and LSTM model.

Conclusion
The EDL model represents a promising step toward integrating physical hydrology with deep learning. However, the manuscript currently understates its methodological limitations and overstates its novelty relative to existing hybrid approaches. Addressing these issues—through expanded validation, methodological transparency, and clearer positioning within the literature— would significantly enhance its contribution to the field.
Recommendation: Major revision required.
References
Feng, D., Liu, J., Lawson, K., Shen, C., 2022. Differentiable, Learnable, Regionalized Process-Based Models With Multiphysical Outputs can Approach State-Of-The-Art Hydrologic Prediction Accuracy. Water Resour. Res. 58, e2022WR032404. https://doi.org/10.1029/2022wr032404
He, L., Shi, L., Song, W., Shen, J., Wang, L., Hu, X., Zha, Y., 2024. Synergizing Intuitive Physics and Big Data in Deep Learning: Can We Obtain Process Insights While Maintaining

State-Of-The-Art Hydrological Prediction Capability? Water Resour. Res. 60, e2024WR037582. https://doi.org/10.1029/2024WR037582

Hoedt, P.-J., Kratzert, F., Klotz, D., Halmich, C., Holzleitner, M., Nearing, G.S., Hochreiter, S., Klambauer, G., 2021. MC-LSTM: Mass-Conserving LSTM, in: Meila, M., Zhang, T. (Eds.), Proceedings of the 38th International Conference on Machine Learning, Proceedings of Machine Learning Research. PMLR, pp. 4275–4286.

Jiang, S., Zheng, Y., Solomatine, D., 2020. Improving AI System Awareness of Geoscience Knowledge: Symbiotic Integration of Physical Approaches and Deep Learning. Geophys. Res. Lett. 47, e2020GL088229. https://doi.org/10.1029/2020gl088229

Li, Z., Kan, G., Yao, C., Liu, Z., Li, Q., Yu, S., 2014. Improved Neural Network Model and Its Application in Hydrological Simulation. J. Hydrol. Eng. 19, 04014019. https://doi.org/10.1061/(ASCE)HE.1943-5584.0000958

Read, J.S., Jia, X., Willard, J., Appling, A.P., Zwart, J.A., Oliver, S.K., Karpatne, A., Hansen, G.J.A., Hanson, P.C., Watkins, W., Steinbach, M., Kumar, V., 2019. Process Guided Deep Learning Predictions of Lake Water Temperature. Water Resour. Res. 55, 9173–9190. https://doi.org/10.1029/2019wr024922

Shen, C., Appling, A.P., Gentine, P., Bandai, T., Gupta, H., Tartakovsky, A., Baity-Jesi, M., Fenicia, F., Kifer, D., Li, L., Liu, X., Ren, W., Zheng, Y., Harman, C.J., Clark, M., Farthing, M., Feng, D., Kumar, P., Aboelyazeed, D., Rahmani, F., Beck, H.E., Bindas, T., Dwivedi, D., Fang, K., Höge, M., Rackauckas, C., Roy, T., Xu, C., Lawson, K., 2023. Differentiable modeling to unify machine learning and physical models and advance Geosciences. Nat. Rev. Earth Environ. 4, 552–567. https://doi.org/10.1038/s43017-023-00450-9

Wang, Y.-H., Gupta, H.V., 2024. Towards Interpretable Physical-Conceptual CatchmentScale Hydrological Modeling Using the Mass-Conserving-Perceptron. Water Resour. Res. 60, e2024WR037224. https://doi.org/10.1029/2024WR037224

Yang, S., Yang, D., Chen, J., Santisirisomboon, J., Lu, W., Zhao, B., 2020. A physical process and machine learning combined hydrological model for daily streamflow simulations of large watersheds with limited observation data. J. Hydrol. 590, 125206.https://doi.org/10.1016/j.jhydrol.2020.125206

Response: Thank you very much for your suggestions, especially for the references you provided, which were extremely helpful in revising our manuscript. In response to the issues you raised, we have made corresponding revisions and improvements in the manuscript, including adding new references, refining the presentation of our innovations, and providing a more detailed explanation of the internal computational process of the proposed XAJRNN neural network unit.

---

## Author Response (AR3)

**Cover Letter**

**Dear Editor and Reviewers**

This is the resubmitted version of the revised manuscript entitled: An explainable deep learning model based on hydrological principles for flood simulation and forecasting (Manuscript No. EGUSPHERE-2025-279). The paper has been revised along the lines suggested by the reviewers. All the comments are addressed in the new version of our paper. We have added a benchmark model and more explanations to the revised manuscript, and modified some tables and figures. The changed and added parts of the text (except some language correction) are marked in RED color for easy review.

It would be greatly appreciated if the revised version of the paper can be re-evaluated by the same reviewer who spent available time to provide constructive and professional comments and suggestions, which have led to improvement on the presentation and quality of the paper.

We sincerely hope you will find the revised version of the paper is to your satisfaction. We are, of course, more than happy to further improve the paper upon request.

In the following, we provide point to point response to the comments of the reviewers.

Yours sincerely,

September 10, 2025

Corresponding author

Prof. Shenglian Guo

State Key Laboratory of Water Resources Engineering and Management,

Wuhan University, Wuhan 430072, P. R China

**E-mail: slguo@whu.edu.cn**

**Response to Reviewers' comments**

Legend

Reviewers' comments

Authors' responses

Direct quotes from the revised manuscript

The authors have submitted a revised version that addresses some, but not all, of the concerns raised in the initial review. The manuscript still lacks the critical analysis of LSTM correction mechanisms that was requested in the first review. This represents a gap in understanding how the proposed EDL model actually functions as a hybrid physics-ML system.

As indicated in Section 3.2.1, two benchmark models are presented: the LSTM model and the XAJ model. While the authors claim in lines 368-369 that " The purpose of this model is to demonstrate the contribution of the XAJRNN layer in the EDL model", a careful examination of the underlying logic reveals a fundamental flaw. Since the primary difference between the EDL model and the XAJ model lies in the LSTM component, comparing EDL against XAJ essentially analyzes the role of the LSTM layer, not the XAJRNN layer as claimed. Conversely, comparing EDL to LSTM would properly demonstrate the contribution of the XAJRNN component.

Moreover, this comparison framework requires first establishing whether XAJRNN outputs differ from XAJ outputs—something that could be verified by extracting and analyzing the Q values directly from the XAJRNN layer. Such analysis would reveal how the XAJRNN functions when combined with an LSTM post-processing layer and how it differs from the directly calibrated XAJ model. Additionally, this study could compare the EDL model with an offline post-processed approach (i.e., using XAJ model outputs as inputs to a separately trained LSTM). This comparison would help readers better understand the value of the online integration of XAJRNN with the post-processing LSTM module.

In summary, the central question is whether the proposed coupling between

physical and ML components provides genuine synergistic benefits or merely represents a sophisticated form of post-processing. This distinction is crucial for understanding the true value of hybrid physics-ML models. Such analysis would elevate this work from a performance comparison to a deeper investigation of physics-ML coupling mechanisms and distinguish it from previous similar research.

Response: We sincerely thank the reviewer for the constructive comments and for highlighting the importance of analyzing the correction mechanism in hybrid physics–ML models. We agree that our previous revision did not sufficiently address this aspect, and we have carefully revised the manuscript to include a more detailed analysis.

We acknowledge the reviewer's observation that comparing EDL with XAJ primarily reflects the role of the LSTM layer, whereas comparing EDL with LSTM better illustrates the contribution of the XAJRNN. We have clarified this logic in the revised manuscript:

Lines 361-363: The first benchmark model is the ordinary XAJ model, which also takes areal mean rainfall and evaporation as input to illustrate the role of the XAJRNN layer in the EDL model.

Lines 372-373: The purpose of LSTM model is to compare the contribution of XAJRNN layer to the simulation performance in the EDL model.

Following the reviewer's suggestion, we have extracted the Q values from the XAJRNN layer and analyzed them against the calibrated XAJ outputs. This analysis has revealed how XAJRNN layer operated when integrated with the LSTM post-processing module and how it differed from a conventional XAJ model.

Lines 413-422: Furthermore, as noted in Section 3.1.2, the XAJRNN layer within the EDL model can directly output simulated outflow ($Q$). To evaluate its performance, we extracted the runoff from the XAJRNN layer and compared it against observed streamflow in these two basins. The results are described as follows: in the Lushui River basin, the training period yielded $NSE$=0.92, $RE$=4.02%, $RMSE$=74.69 m³/s, while the testing period yielded $NSE$=0.90, $RE$=10.87%, $RMSE$=86.98 m³/s. In the Qingjiang River basin, the training period achieved $NSE$=0.89, $RE$=3.84%, $RMSE$=172.64 m³/s,

and the testing period *NSE*=0.86, *RE*=-7.17%, *RMSE*=198.74 m³/s. Compared with the XAJ model, the runoff simulated by the XAJRNN layer shows overall improvement. However, its accuracy still falls short of the full EDL model. These findings confirm that while the XAJRNN layer has advantages over the standard XAJ model, integrating it with the LSTM layer could improve simulation accuracy.

We fully agree that comparing the EDL model with an offline post-processed approach (i.e., using XAJ model outputs as inputs to a separately trained LSTM) would help readers better understand the value of the online integration of XAJRNN with the post-processing LSTM module. We have added a benchmark model (XAJ-LSTM). And subsequent content has been added to compare the performance of the XAJ-LSTM model and the EDL model.

Lines 378-384: The third benchmark model is the XAJ-LSTM hybrid model. This model utilizes the simulated discharge generated by the ordinary XAJ model as its primary input, augmented by observed areal mean rainfall and pan evaporation data. The final output of this model is the simulated flow discharge. Similarly, the training process and hyperparameter configurations for the XAJ-LSTM model are kept consistent with those used in the two previous benchmark models. The purpose of this benchmark model is to demonstrate the superior performance of the proposed EDL model in comparison to using the LSTM layers solely for hydrological post-processing.

Line 434:

Table 2: Comparative analysis of model simulation accuracy evaluation metrics.

| Basin | Model | Training period | | | Test period | | |
|---|---|---|---|---|---|---|---|
| | | NSE | RE (%) | RMSE (m³/s) | NSE | RE (%) | RMSE (m³/s) |
| Lushui River | EDL | 0.98 | 1.59 | 34.11 | 0.98 | -2.69 | 43.71 |
| | XAJ | 0.86 | -26.07 | 93.83 | 0.90 | -18.50 | 89.60 |
| | LSTM | 0.97 | -1.90 | 44.87 | 0.96 | -0.61 | 54.27 |
| | XAJ-LSTM | 0.93 | 4.24 | 70.90 | 0.92 | 19.06 | 73.54 |
| Qingjiang River | EDL | 0.95 | 1.10 | 104.09 | 0.92 | -8.74 | 167.94 |
| | XAJ | 0.85 | 5.91 | 182.05 | 0.85 | -7.92 | 231.17 |
| | LSTM | 0.90 | -4.16 | 147.89 | 0.93 | -6.19 | 155.71 |
| | XAJ-LSTM | 0.88 | 1.56 | 164.52 | 0.86 | -11.80 | 227.62 |

In the revised Discussion section of the manuscript, we have analyzed whether the coupling between the proposed physical and machine learning components provides genuine synergistic benefits from a parameter optimization perspective:

Lines 575-584: Since the parameter adjustments of the hydrological and DL models are conducted independently, the resulting parameter combination is often suboptimal, thereby constraining simulation accuracy. A comparison between the EDL model and the XAJ-LSTM model highlights this issue. The XAJ-LSTM model, which uses the outputs of the traditional XAJ model as inputs and is trained independently, shows some improvement over the XAJ model but still underperforms compared with the EDL model. By contrast, the proposed EDL model integrates hydrological and DL components within a unified framework, enabling synchronized training and joint parameter optimization. This online strategy not only eliminates the parameter mismatch inherent in conventional post-processing methods but also ensures that both hydrological and DL parameters are optimized simultaneously, leading to generate synergistic benefits.

---

## Author Response (AR4)

**Cover Letter**

**Dear Editor and Reviewers**

This is the resubmitted version of the revised manuscript entitled: An explainable deep learning model based on hydrological principles for flood simulation and forecasting (Manuscript No. EGUSPHERE-2025-279). The paper has been revised along the lines suggested by the reviewers. All the comments are addressed in the new version of our paper. We have added more explanations to the revised manuscript. The changed and added parts of the text (except some language correction) are marked in RED color for easy review.

It would be greatly appreciated if the revised version of the paper can be re-evaluated by the same reviewer who spent available time to provide constructive and professional comments and suggestions, which have led to improvement on the presentation and quality of the paper.

We sincerely hope you will find the revised version of the paper is to your satisfaction. We are, of course, more than happy to further improve the paper upon request.

In the following, we provide point to point response to the comments of the reviewers.

Yours sincerely,

November 4, 2025

Corresponding author

Prof. Shenglian Guo

State Key Laboratory of Water Resources Engineering and Management,

Wuhan University, Wuhan 430072, P. R China

**E-mail: slguo@whu.edu.cn**

Legend

Reviewers' comments

Authors' responses

Direct quotes from the revised manuscript

The authors have satisfactorily addressed the comments raised in the previous review round. The manuscript is much improved. I have only a few minor points that require further clarification before the paper can be accepted for publication.

Response: We sincerely appreciate the reviewer's recognition of our revised manuscript. The paper has been improved according to the reviewer's specific comments in the revised version.

Specific Comments:

1. Training Strategy: The study applies the proposed EDL model to different basins. Could the authors please clarify the training strategy used? Specifically, were the model parameters trained jointly using data from all basins simultaneously, or was a separate model trained individually for each basin?

Response: Thank you for your comment. In our study, a separate model was trained for each basin for flood simulation and forecasting. The corresponding clarification has been added to the revised manuscript.

Lines 356-358: In our study, a separate model was trained for each basin for flood simulation and forecasting, whose parameters were directly updated using the standard end-to-end backpropagation approach.

2. Parameter Update Mechanism: The authors state that the parameters in the XAJRNN layer are updated via gradient descent and backpropagation. To be precise, are these parameters updated using a standard, end-to-end backpropagation approach directly? Or is the method more aligned with a differentiable parameter learning (dPL) paradigm, such as the one proposed by Tsai et al. (2021), which is cited in the manuscript?

This clarification is related to point 1. The concern is that if a single set of parameters is optimized via backpropagation for each basin individually, the result is essentially a 'local training' model, similar to traditional calibration. A key benefit of dPL-style approaches is the ability to train 'one model for all basins' (i.e., a regional or global model) by leveraging data from many catchments. Could the authors clarify if their framework is intended to or capable of supporting this more generalizable, multi-basin training approach?

Response: Thank you for your valuable suggestion. As mentioned in our revised manuscript, the model parameters are updated directly using the standard end-to-end backpropagation approach. In our study, a separate set of parameters was optimized for each basin through backpropagation, which essentially results in a "locally training" model, similar to traditional calibration. We have acknowledged this limitation in our study and discussed it. One key advantage of the differentiable

parameter learning approach lies in its ability to leverage data from multiple basins to train a single model applicable to all basins (i.e., a regional or global model). Developing our framework to support this more generalizable multi-basin training approach will be an important direction for future research.

3. Conclusion Expansion: The conclusion section is concise. However, I would suggest the authors briefly and explicitly elaborate on the potential limitations of the current study (e.g., whether the current framework can leverage information from multiple basins as discussed in point 2, or if it is limited to single-basin calibration). Moreover, I recommend a slight expansion to touch upon the broader implications of this work or more specific future research directions that stem from the study's findings and its limitations.

Reference:

Tsai, W. P., Feng, D., Pan, M., Beck, H., Lawson, K., Yang, Y., Liu, J., and Shen, C.: From calibration to parameter learning: Harnessing the scaling effects of big data in geoscientific modeling, Nat. Commun., 12, 5988, https://doi.org/10.1038/s41467-021-26107-z, 2021.

Response: Thank you for your suggestion. Following your advice, we have revised the Discussion section of the paper to discuss the limitations of this study and outline specific directions for future research, as shown below:

Lines 587-599: This study demonstrates that the proposed EDL model, which tightly integrates physical mechanisms with DL, can effectively improve the accuracy of flood simulation and forecasting. A limitation of the present work is that the model parameters were obtained separately for each basin. Consequently, the current implementation represents a locally trained model that is functionally similar to traditional calibration. Future research should therefore pursue two complementary directions to improve generality and adaptability. One is to develop multi-basin (regional or global) training strategies via differentiable parameter learning, transfer learning, or meta-learning, which can leverage data from many basins to produce a single, generalizable model. Another is to introduce mechanisms for dynamic, input-dependent parameter adaptation so that model parameters can evolve with temporal changes in inputs. Additional promising avenues include explicit uncertainty quantification, tighter coupling between physics and data-driven components, and online-updating for real-time forecasting. Pursuing these directions would increase the model applicability across diverse basins and enhance its potential for scientific discovery.